# Eq-WaLa: Equivariant Augmentation and Regularization for Wavelet Latent Flow Matching

**Ka-Hei Hui, Arianna Rampini & Pradyumna Reddy**
Autodesk AI Lab
Autodesk Research
{ka.hei.hui,arianna.rampini,chinthala.pradyumna.reddy}@autodesk.com

**Mehdi Safaee, Aditya Sanghi & Pradeep Kumar Jayaraman**
Autodesk AI Lab
Autodesk Research
{mehdi.safaee,aditya.sanghi,pradeep.kumar.jayaraman}@autodesk.com

## Abstract

Despite their strong performance in compact 3D shape modeling, latent implicit generative models typically neglect geometric symmetries of 3D shapes, particularly rotation invariance, causing representations to depend on arbitrary coordinate frames rather than intrinsic shape structure. To tackle this, we propose an equivariant generative framework for 3D shapes that explicitly accounts for rotational variability during training. Our method builds on a compact wavelet-tree representation of a shape's signed distance field (SDF), enabling multi-scale, channel-efficient encoding and efficient convolutional processing. To address the common challenge of unaligned orientations in real-world datasets, we introduce two key components: (i) a novel wavelet-domain rotation augmentation scheme that enables exact on-the-fly construction of rotated inputs via the inherent equivariance of wavelet transforms, and (ii) a latent-space regularization strategy that enforces consistency under discrete latent rotations. Combined with a flow matching model trained on these augmented and regularized latents, our framework can produce high-quality 3D shapes. We validate the effectiveness of these components, showing improved reconstruction accuracy and faster generative convergence on the Thingi10K dataset, which lacks consistent canonical orientations. Furthermore, we also demonstrate the scalability of our approach to large-scale settings, including both unconditional and text-conditioned shape generation tasks on the complex Objaverse dataset.

## 1 Introduction

3D assets play a vital role in a wide range of applications, including digital content creation for AR/VR, film production, video games, industrial design, and robotic simulation. To support these domains, recent research has explored generative modeling of 3D shapes in various formats, such as voxels, point clouds, and surface meshes. Among these, *implicit functions*, such as occupancy fields or signed distance functions (SDFs), have emerged as a powerful representation for their ability to encode high-resolution geometry, handle complex topologies, and support efficient operations like differentiable rendering and boolean composition.

Despite the representational advantages of implicit functions, modeling them directly at high resolution remains computationally challenging due to the increased dimensionality of the 3D domain. This makes it difficult, if not intractable, for expressive generative models, such as diffusion or autoregressive models, to generate implicit fields end-to-end. To alleviate this, recent works Zhang et al. (2023); Xiong et al. (2024); Ren et al. (2024) have adopted a latent approach, where an autoencoder first learns a compact representation through a reconstruction task. Generative models are then trained to synthesize these low-dimensional latents instead of full-resolution fields, greatly improving scalability and training efficiency.

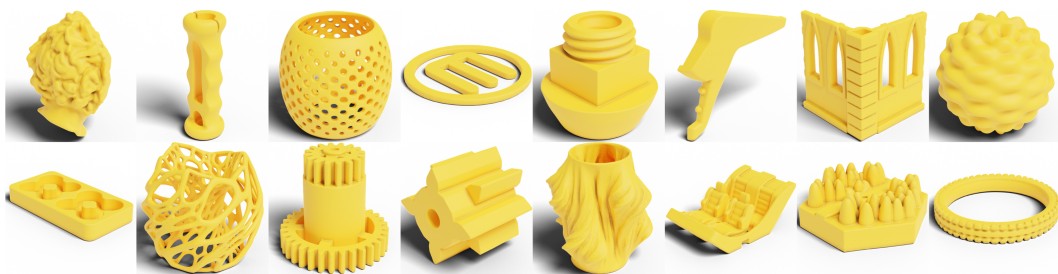

Figure 1: High-quality 3D shapes generated by our model on Thingi10K Zhou & Jacobson (2016). Our method can produce detailed, diverse, and structurally sound outputs across a wide range of categories, demonstrating strong generalization and orientation robustness.

However, most latent generative approaches place less emphasis on geometric symmetries specifically on a fundamental property of 3D shapes like *rotation invariance*. A 3D object can appear in arbitrary orientations, yet its geometry should remain equally valid for generation. Although this omission may not notably affect models trained on canonically aligned datasets like ShapeNet Chang et al. (2015), it presents substantial potential for boosting generation performance on datasets where objects exhibit diverse and unaligned orientations. Leveraging this property, we propose an equivariant latent implicit generative framework that explicitly accounts for the discrete rotations of 3D shapes during training, improving generalization and robustness across orientations.

Our framework builds on the wavelet-tree representation Hui et al. (2024); Sanghi et al. (2024), which encodes a 3D shape's truncated signed distance field (TSDF) into a compact, multi-scale, and channel-efficient volume. This representation enables accurate reconstruction while remaining highly compatible with convolutional neural networks due to its regular volumetric structure. We first train a latent autoencoder that maps wavelet-tree volumes into a compact latent space, which can later be modeled by a generative model.

The key insight of our approach is that both the wavelet-tree volume and its latent volume are defined on regular voxel grids, allowing us to apply exact discrete (90-degree) 3D rotations, which are common in 3D modeling. We exploit this in two complementary ways: (i) through *wavelet-domain rotation augmentation*, we construct rotated training inputs on the fly by leveraging the equivariant structure of wavelet transforms; and (ii) through *equivariant regularization*, we enforce consistency in the latent space by supervising decoded outputs under randomly applied latent rotations. Together, we show that these enable learning a high-quality, orientation-consistent latent space, which we combine with a flow-matching model for robust 3D shape generation on an unaligned dataset (as shown in Figure 1).

Our main contributions are summarized as follows:

- We introduce an efficient method for applying discrete 3D rotations directly in the wavelet domain during autoencoder training. This enables on-the-fly augmentation without precomputed samples, resulting in improved reconstruction quality.

- We propose a novel latent-space regularization technique that enforces rotation-consistent decoding, improving generation quality and accelerating convergence of the generative model without relying on equivariant network architectures.

- We demonstrate that our augmented and regularized latent space enables a flow matching model to generate diverse, high-fidelity 3D shapes on unaligned datasets like Thingi10K Zhou & Jacobson (2016).

- We extend our framework to large-scale settings by demonstrating its scalability for both unconditional and text-conditioned generation on the Objaverse dataset.

## 2 RELATED WORK

**3D Implicit Generation.** Among different 3D representations, implicit representations have gained popularity for their flexibility and ability to capture fine geometric details. However, generating high-resolution implicit fields remains computationally challenging. To mitigate this, existing methods

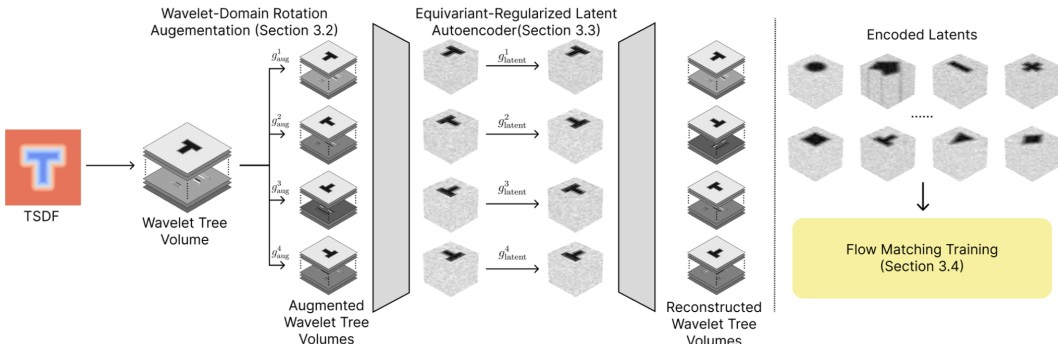

Figure 2: Framework overview. We convert TSDFs to compact wavelet tree volumes, apply exact wavelet-domain rotations, train an equivariantly regularized latent autoencoder, and learn a flow-matching model in latent space (Sections 3.1–3.4).

often generate coarse implicit volumes Zheng et al. (2023), apply lossy compression Hui et al. (2022; 2024); Zhou et al. (2024), or learn a low-dimensional latent representation Sanghi et al. (2024); Zhang et al. (2023); Xiong et al. (2024); Ren et al. (2024); Petrov et al. (2024); Zhang & Wonka (2024). Our approach adopts the latent modeling paradigm but diverges by training a flow matching model within a regularized, equivariant latent space to enhance generative performance.

**3D Rotation Equivariant Models.** Many works have leveraged the 3D shapes' rotation-invariant property to design rotation-equivariant neural networks for tasks such as classification, segmentation, and reconstruction. Approaches include PCA-based normalization Yu & Sun (2024), spherical harmonics Esteves et al. (2018); Thomas et al. (2018); Fuchs et al. (2020), equivariant graph networks Satorras et al. (2021); Huang et al. (2022); Schütt et al. (2021), and vector neuron models Chen et al. (2024); Deng et al. (2021); Park et al. (2024). In generative modeling, equivariance has been explored in molecular domains, where models aim to assign equal probabilities to rotated instances Garcia Satorras et al. (2021); Midgley et al. (2023); Hoogeboom et al. (2022); Xu et al. (2022); Song et al. (2023). In contrast to these works, our method does not enforce strict architectural equivariance, which could be limiting to generation performance. Instead, we introduce a lightweight regularization strategy, encouraging the latent space to be consistent across rotations.

**Latent Space Regularization.** Latent regularization is critical in 2D generative models to prevent mode collapse or excessive variance in the latent distributions. Classical approaches address this by enforcing distributional constraints using KL divergence Kingma et al. (2013) or vector quantization Van Den Oord et al. (2017). However, recent studies Skorokhodov et al. (2025); Kouzelis et al. (2025); Yao et al. (2025) show that these strategies can be insufficient, especially for preserving fidelity in downstream generation tasks. To resolve this, they introduce semantic priors from pretrained vision-language models Yao et al. (2025) or impose equivariance constraints in the latent space Skorokhodov et al. (2025); Kouzelis et al. (2025) to improve generative robustness. While these ideas have been explored in 2D image generation, they remain under-explored in 3D. Our work is, to our knowledge, the first to apply equivariant latent regularization to 3D latent generative models, showing its effectiveness in combination with wavelet representations and flow-based models.

**Steerable Wavelet Filters.** Steerable filters and wavelet-like pyramids provide a principled way to handle rotations by expressing a rotated filter (or its response) as a linear combination of a small set of basis filters, with combination weights determined solely by the rotation Freeman et al. (1991). This concept underlies multi-scale oriented decompositions such as the steerable pyramid Simoncelli et al. (1995), and has been formalized more broadly via steerable wavelet frames, where rotations act through low-dimensional steering matrices that mix orientation channels while preserving perfect reconstruction Unser & Van De Ville (2009); Unser et al. (2011); Chenouard & Unser (2012). In our setting, we do not redesign the wavelet basis to obtain continuous-angle steerability; instead, we exploit the induced action of the finite cube-rotation group on a compact 3D wavelet tree volume. For the discrete rotations we consider and a suitable separable wavelet basis, this yields an exact and highly efficient instance of steerability. Therefore, our on-the-fly wavelet-domain rotation augmentation can be viewed as a concrete specialization of steerable wavelet principles to wavelet tree representations that are well-suited for latent 3D generation.

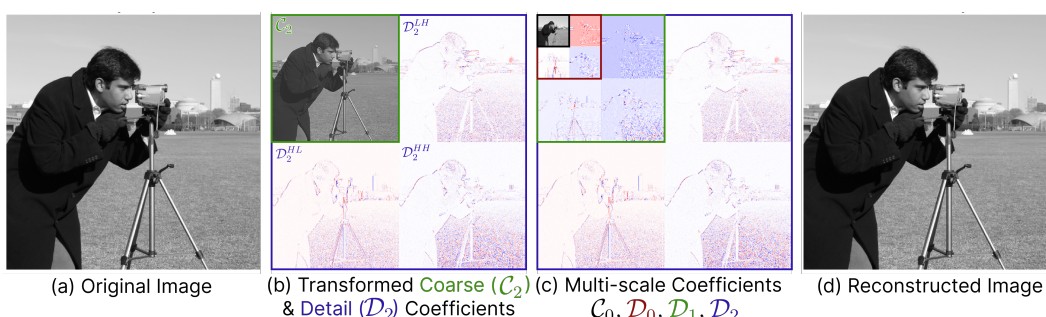

Figure 3: Starting from an image (a), a wavelet transform produces one coarse and three detail subband coefficients (b). Recursively decomposing the coarse subband yields multi-scale coefficients (c), enabling lossless reconstruction of the original image (d).

## 3 METHOD

Figure 2 illustrates our equivariant latent generative framework. We first convert the input TSDF into a compact, multi-scale wavelet tree volume representation (Section 3.1). To efficiently model rotational invariance, we introduce an exact method for applying discrete 3D rotations directly in the wavelet domain, enabling on-the-fly augmentation during autoencoder training without recomputation (Section 3.2). To promote latent consistency across orientations, we further introduce an equivariant regularization strategy applied during training (Section 3.3). Finally, we train a flow-based generative model over the latent space to enable scalable and diverse 3D shape synthesis (Section 3.4).

### 3.1 PRELIMINARY

**Wavelet Decomposition.** We begin with an overview of wavelet decomposition using a 2D example in Figure 3. Given an image $\mathcal{I}$, a wavelet transform $\phi$ decomposes it into coarse and detail components: $\phi(\mathcal{I}) = \{\mathcal{C}_2, \mathcal{D}_2\}$. In 2D, $\mathcal{D}_2$ consists of three subbands: $\mathcal{D}_2 = \{\mathcal{D}_2^{LH}, \mathcal{D}_2^{HL}, \mathcal{D}_2^{HH}\}$ (Figure 3 (b)). By recursively applying $\phi$ to the coarse component, we obtain a multi-scale representation consisting of a final coarse band $\mathcal{C}_0$ and detail bands $\{\mathcal{D}_0, \ldots, \mathcal{D}_{N-1}\}$ (Figure 3 (c)). The representation is lossless and enables exact reconstruction of the image, as shown in Figure 3 (d).

**Wavelet Tree Volume Representation.** We represent 3D shapes using a compact wavelet tree volume Hui et al. (2024). Specifically, for each 3D shape $S$, we first sample a truncated signed distance field (TSDF) $\Phi(\mathbf{x})$ at a resolution of $256^3$. We then apply the above-mentioned procedure to obtain a set of multi-scale wavelet coefficients. By filtering out insignificant coefficients and stacking different subbands in the channel dimension, we construct a wavelet tree volume, which can be used to reconstruct the TSDF volume via inverse wavelet transform. We denote this procedure as $\mathcal{V}(\mathcal{S}) \in \mathbb{R}^{d^3 \times C}$, where $d$ is the spatial resolution and $C$ is the channel size. In our setting, the volume has a reduced spatial resolution ($d = 48$), while different subbands are stacked along the channels ($C = 64$). This compact representation preserves fine geometric details while maintaining a regular, low-resolution spatial structure that is well-suited for convolution-based neural networks. For procedural details, please refer to the Appendix or Hui et al. (2024).

**Wavelet Latent Representation.** Although the wavelet tree volume is already a compact representation, we follow Sanghi et al. (2024) and introduce a latent autoencoder for improved efficiency. An encoder $E(\cdot)$ maps the input $\mathcal{V}(S)$ to a latent volume $z \in \mathbb{R}^{d'^3 \times C'}$ with reduced spatial and channel dimensions ($d' = 12, C' = 8$), and a decoder $D(\cdot)$ reconstructs the volume as $\mathcal{V}'(S)$, which can produce a reconstructed TSDF $\Phi'(\mathbf{x})$. The generative model only synthesizes $z$, greatly simplifying learning and generation.

**Wavelet Tree Volumes of Rotated Shapes.** A key property of 3D shapes is that their underlying structure $\mathcal{S}$ remains unchanged under rotation transformations $\mathcal{T}(\mathcal{S})$, where $\mathcal{T}$ denotes a rotation applied to the shape. Since rotated shapes are equally valid from a generative perspective, it is essential for our latent autoencoder to reconstruct shapes across all orientations; that is, the decoder should satisfy $D(E(\mathcal{V}(\mathcal{T}(S)))) \simeq \mathcal{V}(\mathcal{T}(\mathcal{S}))$. A typical approach, following Sanghi et al. (2024), is to precompute and store a subset of rotated wavelet tree volumes $\mathcal{V}(\mathcal{T}(S))$ for data augmentation.

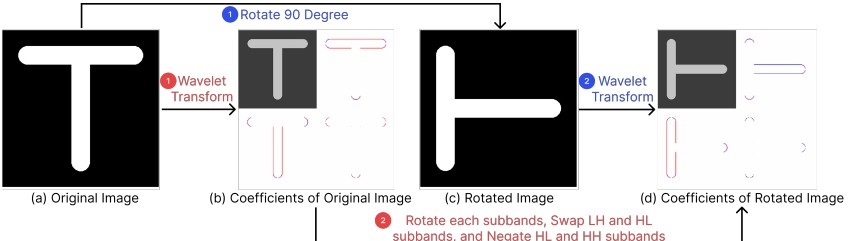

Figure 4: Given an original image (a), one can either rotate the image (c) and compute its wavelet coefficients (d), or directly transform wavelet coefficients of the original image (b) by rotating each subband, swapping $LH$ and $HL$, and negating $HL$ and $HH$. Both approaches yield identical results.

However, this strategy becomes inefficient when considering the large number of possible rotations, *e.g.*, 24, even with only 90-degree rotations, leading to significant preprocessing and storage overhead.

## 3.2 WAVELET-DOMAIN ROTATION AUGMENTATION

**Equivariance in Wavelet Transforms.** We begin by formalizing equivariance with respect to finite discrete rotation groups, such as the 2D dihedral group $D_4$ and the 3D octahedral group $O$. Let $f : U \rightarrow W$ be a transformation between vector spaces. We say that $f$ is equivariant to a group $G$ if the following condition holds:

$$f\big(\rho_{\text{in}}(g) \cdot x\big) = \rho_{\text{out}}(g) \cdot f(x), \quad \forall g \in G, \ x \in U, \tag{1}$$

where $\rho_{\text{in}}(g)$ and $\rho_{\text{out}}(g)$ are group actions, *i.e.*, representations of $g$, on the input and output spaces, respectively. In our context, $\rho_{\text{in}}(g)$ corresponds to a discrete spatial rotation $\mathcal{T}(g)$ applied to a 2D image or 3D volume, and $\rho_{\text{out}}(g)$ defines the induced transformation on the wavelet coefficients. If we can explicitly characterize the output transformation $\rho_{\text{out}}(g)$ for a given wavelet transform $\phi$, we can use the equivariance relation above to obtain the wavelet coefficients of a rotated input without recomputing the transform from scratch.

The key observation is that, with a suitably chosen wavelet basis, such as the biorthogonal `bior3.9` filter, certain symmetry properties emerge in the coefficients under spatial rotations. To illustrate this, we present a 2D example in Figure 4. Given an input image (a), one can either:

- rotate the image (c) and compute its wavelet coefficients (d), or
- compute the wavelet coefficients from the original image (b) and apply a transformation directly to the coefficients.

By comparing the transformed coefficients in both approaches, we observe a strategy effectively mimics the rotated coefficients using the following operations:

1. **Spatial rotation** of each subband,
2. **Negation** of the $HL$ and $HH$ subbands,
3. **Swapping** of the $LH$ and $HL$ subbands.

This transformation defines the output group action $\rho_{\text{out}}(g)$ and confirms the equivariant structure of the wavelet transform under discrete rotations. The result matches the coefficients of the rotated image, effectively defining $\rho_{\text{out}}(g)$.

**Equivariance in 3D Wavelet Volumes.** We extend this principle to the 3D wavelet tree volume representation $\mathcal{V}(S)$. As subbands in 3D are organized along the channel dimension, the equivariant transformation corresponding to a rotation $\mathcal{T}(g)$ consists of:

1. Applying a spatial rotation to the entire volume,
2. Negating specific channels (subbands) based on the rotation,
3. Permuting the wavelet channels according to the induced subband reordering.

This enables us to compute $\mathcal{V}(\mathcal{T}(S))$ from $\mathcal{V}(S)$, without reapplying the wavelet transform, thereby enabling efficient rotation augmentation. Details of channel mappings are provided in the Appendix.

**Why This Matters.** Unlike Sanghi et al. (2024), our wavelet-domain rotation augmentation only requires three lightweight tensor operations and supports on-the-fly generation of all discrete 3D rotations, eliminating additional preprocessing or storage. In Section 4.5, we demonstrate that training the latent autoencoder with rotation-augmented input leads to consistent improvements on unaligned data, highlighting the effectiveness of our augmentation strategy.

### 3.3 EQUIVARIANTLY REGULARIZED LATENT AUTOENCODER

**Rotation-Augmented Autoencoder Training.** To improve generalization across different orientations, we train the autoencoder using wavelet-domain rotation augmentation. For each shape $S$, we randomly sample a group element $g_{\text{aug}} \in G$ and construct the rotated wavelet tree volume $\mathcal{V}(\mathcal{T}_{g_{\text{aug}}}(S))$ using the method described in Section 3.2. This augmented volume is then encoded via $E(\cdot)$ to obtain a latent volume $z$, and decoded by $D(\cdot)$ to produce a reconstructed wavelet tree volume $\mathcal{V}'$. To supervise this process, we adopt the adaptive sampling loss $L_{\text{recon}}(\mathcal{V}', \mathcal{V})$ from Hui et al. (2024), which prioritizes high-magnitude wavelet coefficients while maintaining coverage through random sampling. Further implementation details are provided in the Appendix.

To further improve reconstruction accuracy near the surface, we introduce additional losses using the reconstructed TSDF $\Phi'(\mathbf{x})$:

$$L_{\text{sdf}}(\mathcal{V}') = \|\Phi'(\mathbf{x})\|_2^2, \tag{2}$$

$$L_{\text{eik}}(\mathcal{V}') = \|\nabla\Phi'(\mathbf{x}) - \mathbf{1}\|_2^2, \tag{3}$$

$$L_{\text{sn}}(\mathcal{V}') = 1 - \left\langle \frac{\nabla\Phi'(\mathbf{x})}{\|\nabla\Phi'(\mathbf{x})\|}, \hat{n} \right\rangle, \tag{4}$$

where $L_{\text{sdf}}$ encourages near-zero TSDF values at surfaces, $L_{\text{eik}}$ enforces unit gradient norms for SDF behavior, and $L_{\text{sn}}$ aligns TSDF gradients with ground-truth normals for geometric accuracy.

We also apply KL regularization on $z$ to align it with a standard Gaussian prior:

$$L_{\text{kl}}(z) = \text{KL}\left(q(z)\|\mathcal{N}(0, I)\right). \tag{5}$$

The overall loss for training the autoencoder without equivariant regularization becomes:

$$L = L_{\text{recon}} + \lambda_{\text{sdf}}L_{\text{sdf}} + \lambda_{\text{eik}}L_{\text{eik}} + \lambda_{\text{sn}}L_{\text{sn}} + \lambda_{\text{kl}}L_{\text{kl}}. \tag{6}$$

**Equivariance Regularization.** To further enforce latent consistency under rotation, we introduce an equivariant regularization strategy that applies multiple group transformations both in the input and latent spaces. For each training shape $S$, we first sample $N$ rotation elements $\{g_{\text{aug}}^1, \ldots, g_{\text{aug}}^N\} \in G$ and generate the corresponding augmented wavelet volumes $\mathcal{V}(\mathcal{T}_{g_{\text{aug}}^n}(S))$ for $n = 1, \ldots, N$. We then encode each volume to obtain latent volumes $z^n = E(\mathcal{V}(\mathcal{T}_{g_{\text{aug}}^n}(S)))$. For each $z^n$, we sample another rotation $g_{\text{latent}}^n \in G$ and apply it to obtain transformed latents $\mathcal{T}_{g_{\text{latent}}^n}(z^n)$, which are decoded as:

$$\hat{\mathcal{V}}^n = D(\mathcal{T}_{g_{\text{latent}}^n}(z^n)). \tag{7}$$

To supervise these outputs, we construct ground-truth volumes on-the-fly as $\mathcal{V}(\mathcal{T}_{g_{\text{latent}}^n}(\mathcal{T}_{g_{\text{aug}}^n}(S)))$. We define the training loss by averaging over all $N$ pairs of input and latent-space rotations:

$$L_{\text{avg}} = \frac{1}{N}\sum_{n=1}^{N}\Big[L_{\text{recon}}(\hat{\mathcal{V}}^n, \mathcal{V}(\mathcal{T}_{g_{\text{latent}}^n}(\mathcal{T}_{g_{\text{aug}}^n}(S)))) + \lambda_{\text{sdf}}L_{\text{sdf}}(\hat{\mathcal{V}}^n) +$$
$$\lambda_{\text{eik}}L_{\text{eik}}(\hat{\mathcal{V}}^n) + \lambda_{\text{sn}}L_{\text{sn}}(\hat{\mathcal{V}}^n)\Big]. \tag{8}$$

**Final Training Objective.** When equivariant regularization is applied, the final training loss becomes:

$$L = L_{\text{avg}} + \lambda_{\text{kl}} \cdot \frac{1}{N}\sum_{n=1}^{N}L_{\text{kl}}(z^n). \tag{9}$$

This formulation can encourage the decoder to produce accurate and consistent reconstructions under latent rotations, while also promoting a well-regularized latent distribution. Additional visualizations of decoded shapes from rotated latents, provided in the Appendix, further demonstrate the effectiveness of this approach.

### 3.4 LATENT FLOW MATCHING

**Flow Matching in Latent Space.** To support fast and scalable sampling, we train a continuous normalizing flow (CNF) model Chen et al. (2018) over the latent space using flow matching Lipman et al. (2022). Traditional CNF models evolve a base distribution $q_0(z_0)$ (typically Gaussian) into a target distribution $q_1(z_1)$ using a time-dependent vector field $v_{\theta,t}$ governed by an ordinary differential equation (ODE):

$$\frac{d}{dt} z_t = v_{\theta,t}(z_t), \quad z_0 \sim q_0. \tag{10}$$

However, training such models requires ODE integration, which is computationally expensive.

Flow matching provides a more efficient alternative by avoiding simulation. It directly supervises the flow field using a simple path-wise objective. We sample $z_0 \sim q_0$ and $z_1 \sim q_1$, and define a linear interpolation path $z_t = (1-t)z_0 + tz_1$. The target velocity is defined as $u_t(z_t \mid z_1) = z_1 - z_0$. The model is then trained using the conditional flow matching (CFM) loss:

$$L_{\text{CFM}} = \mathbb{E}_{t, q_0(z_0), q_1(z_1)} \left\| v_{\theta,t}(z_t) - (z_1 - z_0) \right\|_2^2, \tag{11}$$

which encourages $v_{\theta,t}$ to learn the transport direction between the base and target latent distributions.

## 4 EXPERIMENT

### 4.1 EXPERIMENT SETUP

**Dataset.** To investigate the importance of equivariance in 3D generative modeling, we adopt the Thingi10K dataset Zhou & Jacobson (2016), which contains 10,000 high-quality, 3D-printing-ready models collected from Thingiverse. Unlike canonical datasets such as ShapeNet Chang et al. (2015), which consists of a limited number of object categories with consistently aligned shapes within each category, Thingi10K provides a more diverse and realistic setting. Its models are crowd-sourced from various creators and span a wide range of categories, including mechanical parts, household items, toys, and figurines. Because these objects lack standardized orientation, the dataset naturally contains unaligned shapes, making it well-suited for evaluating the robustness and equivariance of 3D generative models under real-world conditions.

**Baselines.** We compare our equivariant generative framework against six existing methods for implicit 3D shape generation: LAS-Diffusion Zheng et al. (2023), UDiff Zhou et al. (2024), WaveGen Hui et al. (2022), Make-A-Shape Hui et al. (2024), OctFusion Xiong et al. (2024), WaLa Sanghi et al. (2024), Shape2VecSet Zhang et al. (2023), LaGeM Zhang & Wonka (2024), and GEM3D Petrov et al. (2024). Among these, LAS-Diffusion, UDiff, and WaveGen generate coarse or lossy implicit representations directly, OctFusion, WaLa, Shape2VecSet, LaGeM, and GEM3D follow a two-stage pipeline with a latent autoencoder and generative model, while Make-A-Shape bypasses the latent stage by directly generating wavelet-tree volumes from noises. For fair comparison, we retrain all models using their official implementations on the Thingi10K training split, including both the generative and autoencoder components where applicable. Note that we only compare LaGeM on its reconstruction performance, as its generator training code is unavailable.

**Implementation Details.** Our autoencoder is a fully convolutional network with 24 downsampling and 24 upsampling (transpose convolution) layers. We use a 3D adaptation of DiT Peebles & Xie (2023) as our generator, with a small variant (8 attention layers) matching prior methods, and a large variant (48 layers) matching WaLa's capacity. The autoencoder is trained for 400,000 iterations (batch size 8, learning rate $5\mathrm{e}{-5}$), and the generators are trained with batch size 128 and learning rate $2\mathrm{e}{-4}$, 400,000 iterations for the small and 200,000 for the large variant. We use EMA (decay 0.999) during training, employing 8 H100 GPUs for the autoencoder (5 days) and 32 H100 GPUs for the large generator (2 days). We sample four discrete rotation elements ($N = 4$) during training and use fixed loss weights across all experiments, set to $\lambda_{\text{sdf}}, \lambda_{\text{eik}}, \lambda_{\text{sn}}, \lambda_{\text{kl}} = 1.0, 0.1, 1.0, 10^{-4}$.

### 4.2 3D SHAPE AUTOENCODING

**Evaluation Metrics.** To assess our model's 3D autoencoding performance against baseline methods, we use three standard shape comparison metrics: (i) Intersection over Union (IoU) quantifies the volumetric overlap by computing the ratio between the intersection and the union of voxelized shapes; (ii) Light Field Distance (LFD) Chen et al. (2003) evaluates visual similarity by comparing sets of images rendered from multiple viewpoints; and (iii) Chamfer Distance (CD) measures the average

Table 1: Autoencoding results. Our method achieves the highest IoU and lowest CD and LFD. CD is in $\times 10^{-3}$.

| Method | IoU ↑ | CD ↓ | LFD ↓ |
|---|---|---|---|
| LAS-Diff | 70.88 | 2.419 | 2279.6 |
| UDiff | 81.98 | 1.490 | 1431.9 |
| WaveGen | 87.56 | 1.324 | 1263.7 |
| OctFusion | 90.53 | 1.341 | 1139.5 |
| WaLa | 92.96 | 1.220 | 1068.8 |
| Shape2VecSet | 59.71 | 3.235 | 3246.0 |
| LaGeM | 72.82 | 1.934 | 1698.7 |
| GEM3D | 82.34 | 1.613 | 1654.6 |
| Ours | **97.71** | **1.199** | **897.7** |

Table 2: Unconditional 3D shape generation results. Our method outperforms baselines across all metrics, showing better diversity (COV), fidelity (MMD), distributional alignment (1-NNA), and visual quality (FID, KID). CD and EMD are both reported where applicable. Best results in red, second-best in blue.

| Method | COV ↑ | | MMD ↓ | | 1-NNA (~50%) | | FID ↓ | KID ↓ |
|---|---|---|---|---|---|---|---|---|
| | CD | EMD | CD | EMD | CD | EMD | | |
| LAS-Diff | 40.28 | 44.45 | 2.018 | 0.947 | 69.31 | 64.47 | 59.40 | 2.745 |
| UDiff | 34.20 | 39.78 | 2.222 | 0.989 | 81.07 | 74.04 | 58.02 | 2.512 |
| WaveGen | 34.87 | 39.49 | 2.133 | 1.004 | 76.64 | 73.03 | 84.61 | 4.018 |
| OctFusion | 31.70 | 38.80 | 2.357 | 1.041 | 73.97 | 69.45 | 96.88 | 4.878 |
| Shape2Vec | 20.74 | 34.10 | 2.315 | 1.080 | 88.70 | 85.10 | 86.99 | 4.268 |
| WaLa | 37.17 | 42.54 | 2.112 | 0.978 | 72.83 | 67.91 | 53.15 | 1.883 |
| Make-A-Shape | 40.81 | 46.84 | 2.396 | 1.103 | 68.24 | 67.23 | 59.82 | 2.717 |
| GEM3D | 17.53 | 24.77 | 2.992 | 1.210 | 89.75 | 83.69 | 156.28 | 9.657 |
| Ours (Small) | 42.08 | 47.61 | 1.977 | 0.943 | 67.12 | 61.80 | 32.67 | 0.937 |
| Ours (Large) | 42.74 | 47.31 | 1.664 | 0.803 | 67.30 | 62.50 | 21.56 | 0.364 |

closest-point distance between two point clouds sampled from the predicted and ground-truth surfaces in both directions. We compute these metrics on the Thingi10K validation set. A higher IoU and lower LFD and CD reflect better reconstruction quality.

**Quantitative Comparisons.** Table 1 presents the quantitative results compared to baseline methods. For LAS-Diffusion, UDiff, and WaveGen, we evaluate meshes extracted using Marching Cubes from their coarse or lossy representations against the ground-truth shapes. Overall, our method achieves the highest IoU and the lowest values for both LFD and CD, demonstrating superior reconstruction quality. These results indicate that our approach provides the best autoencoding performance among all baselines, offering a stronger foundation for generative model training.

**Qualitative Comparisons.** We present a visual comparison of reconstruction results produced by our method and the baselines in the Appendix. Our model consistently reconstructs shapes with high fidelity, preserving intricate surface geometry and complex topological structures.

### 4.3 UNCONDITIONAL SHAPE GENERATION

**Evaluation Metrics.** We evaluate unconditional shape generation using three geometry-based and two rendering-based metrics. (i) Minimum Matching Distance (MMD) measures fidelity by averaging distances between generated shapes and their nearest ground-truth counterparts; (ii) Coverage (COV) quantifies diversity by checking how many ground-truth shapes are matched by the generated set; (iii) 1-Nearest Neighbor Accuracy (1-NNA) evaluates how distinguishable generated shapes are from real ones; (iv) Fréchet Inception Distance (FID) and (v) Kernel Inception Distance (KID) assess visual realism using image embeddings from rendered views. Lower MMD, FID, and KID, along with higher COV and 1-NNA near 50%, indicate better generative quality. Geometric metrics are computed using both CD and EMD for a comprehensive evaluation.

**Quantitative Comparisons.** Table 2 presents the results for unconditional 3D shape generation across five metrics. Our small generator already outperforms all baselines, achieving the highest COV and lowest MMD (under both CD and EMD), reflecting strong fidelity and diversity. It also reaches 1-NNA values closest to 50% and obtains the lowest FID and KID, indicating superior distributional alignment and visual quality. With the larger model size matching WaLa, our method further improves rendering metrics, demonstrating scalability and effectiveness across settings.

**Qualitative Comparisons.** We provide visual comparisons of uncurated samples generated by our method and the baseline generative models in the Appendix. Our model consistently produces a greater number of shapes with fine-grained geometric details, complex topologies, and higher overall diversity in structure and category.

**Framework Scalability.** To evaluate the scalability of our approach, we extend it to a large-scale setting using a subset of the Objaverse dataset Deitke et al. (2022; 2023), filtered following Xiang et al. (2024). Despite the increased data volume and diversity, our framework generalizes well, synthesizing complex and high-quality 3D shapes across varied categories. Qualitative results on the Objaverse are provided in the Appendix.

### 4.4 TEXT-CONDITIONED SHAPE GENERATION

A visual gallery of text-to-shape generation results is provided in the Appendix, along with comparisons against TRELLIS Xiang et al. (2024) and XCube Zheng et al. (2023). Our method generates

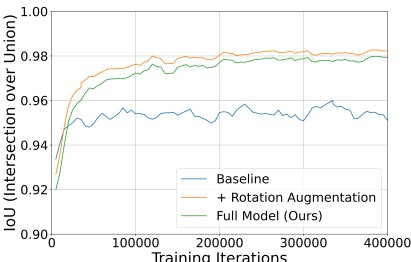

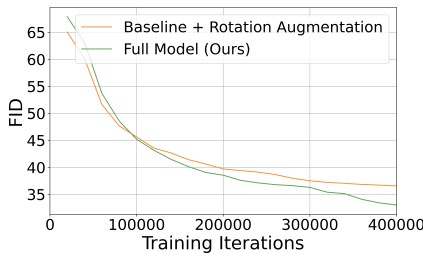

Figure 5: Validation IoU over training iterations for the baseline, with rotation augmentation, and the full model with equivariant regularization. Rotation augmentation yields clear gains, while equivariant regularization introduces only a minor trade-off in reconstruction quality.

Figure 6: Validation FID over training iterations for the model with rotation augmentation and our full model with equivariant regularization. The full model achieves better generative performance overall and reaches the same FID as the augmented baseline (FID = 36) with significantly fewer iterations (240k v.s. 400k), demonstrating improved training efficiency.

shapes that are semantically aligned with the input captions while maintaining coherent geometry and fine-grained structure. Across diverse prompts, the results are broadly comparable to TRELLIS and generally exhibit cleaner, more stable geometry than XCube. These examples demonstrate the potential of our equivariant framework for controllable 3D synthesis, suggesting promising directions for future work in multimodal generation.

## 4.5   ABLATION STUDY

We present ablations analyzing how wavelet-domain rotation augmentation and equivariant regularization each contribute to reconstruction and generation quality.

**Experimental Settings.** We evaluate three training configurations to isolate the effects of each component: (1) *Baseline*: an autoencoder trained without any rotation augmentation or equivariant regularization. (2) *+ Rotation Augmentation*: the baseline model enhanced with wavelet-domain rotation augmentation during training. (3) *Full Model*: the complete version of our method, incorporating both rotation augmentation and equivariant latent regularization. For efficiency, all evaluations are performed using the small generator variant. To ensure a fair comparison, we increase the baseline batch size by a factor of four, matching the sample count when rotation elements are introduced.

**Reconstruction Performance.** Figure 5 illustrates the impact of wavelet-domain rotation augmentation and equivariant regularization on reconstruction performance over training. Compared to the baseline, introducing rotation augmentation boosts validation IoU across most iterations, demonstrating its effectiveness in improving robustness to orientation. Adding equivariant regularization results in a marginal reduction in peak IoU but maintains strong overall performance, validating its utility for achieving latent consistency without significantly compromising reconstruction.

**Generation Performance.** Figure 6 shows the evolution of validation FID during training with and without equivariant regularization. While the addition of regularization introduces a slight reduction in reconstruction performance (as seen in Figure 5), it leads to improved generative quality over the course of training. In particular, it achieves better generative quality at convergence and reaches comparable FID to the augmented baseline much earlier, around 1.6x speed up (240,000 iterations compared to 400,000). This suggests that equivariant regularization not only improves the final generative performance but also accelerates convergence.

## 5   CONCLUSION

Overall, we introduced an equivariant generative framework for 3D shapes that explicitly models rotational variability during training. We leverage a compact wavelet-tree SDF representation and contributes (i) an efficient wavelet-domain rotation augmentation strategy and (ii) an equivariant latent regularization. Together, these components yield an orientation-robust latent space, which we model with a flow-matching generator. Extensive experiments on Thingi10K validate strong reconstruction and generation performance, with additional scalability demonstrated on Objaverse for both unconditional and text-conditioned synthesis.

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

## A    WAVELET-TREE VOLUME CONSTRUCTION

To represent 3D shapes compactly while preserving high fidelity, we adopt the *wavelet-tree representation* introduced in Make-A-Shape Hui et al. (2024) and further employed in WaLa Sanghi et al. (2024). This representation encodes the truncated signed distance field (TSDF) of a shape into a structured, multi-scale volume through 3D wavelet decomposition.

**Wavelet Decomposition.**    Given a 3D shape, we first compute its TSDF on a $256^3$ grid, normalized to lie within the unit cube $[-0.9, 0.9]^3$. A 3D wavelet transform is then applied to the TSDF using a biorthogonal wavelet basis, *i.e.*, `bior3.9`. The transform recursively decomposes the volume into a coarse coefficient volume $C_0$ and detail coefficients $\{D_0, D_1, D_2\}$, where each $D_i$ contains seven directional subbands capturing high-frequency information along combinations of spatial axes. This produces a tree-like structure, where $C_0$ represents the low-frequency (global) content, and $D_i$ encode increasingly localized high-frequency content at finer scales.

**Coefficient Filtering.**    To achieve compactness while retaining reconstruction fidelity, we discard the highest-frequency detail coefficients $D_2$. Among the remaining detail coefficients, we identify a sparse subset of spatially significant entries from $D_0$ using a coefficient filtering strategy:

- For each spatial position $v$ in $D_0$, we compute the maximum absolute coefficient across its 7 subbands: $s(v) = \max_{b=1,\ldots,7} |D_0(v, b)|$.

- We retain the top-$K$ entries (typically $K = 16384$) with the highest scores $s(v)$ as the set of informative detail locations $\mathcal{V}_{D_0}$.

- For each selected spatial position $v \in \mathcal{V}_{D_0}$, we also retain its corresponding $2 \times 2 \times 2$ child region in $D_1$, located at positions $[2v : 2v + 1]$.

**Volume Packing.**    To enable compatibility with convolutional networks, the filtered coefficients are packed into a dense tensor of size $48^3 \times 64$. The packing proceeds as follows:

- The coarse volume $C_0 \in \mathbb{R}^{48^3 \times 1}$ is stored in one channel.

- The selected $D_0$ subbands at positions $\mathcal{V}_{D_0}$ are stored in 7 channels.

- These child coefficients from $D_1$ are aggregated into $7 \times 2^3 = 56$ additional channels, preserving multi-scale directional information at finer resolution. We follow a fixed ordering of the $2 \times 2 \times 2$ children based on the $z \rightarrow y \rightarrow x$ axis traversal.

- Empty entries in all channels are zero-padded.

The resulting packed tensor, referred to as the *wavelet-tree volume*, is a compact and nearly lossless representation of the original TSDF that supports exact reconstruction via inverse wavelet transform. Its regular voxel grid structure enables efficient processing via 3D convolutions and facilitates fast on-the-fly data augmentation, including spatial rotations as discussed in Section 3.2.

## B    ADAPTIVE SAMPLING LOSS

To effectively train the autoencoder on wavelet-tree representations, we adopt the *adaptive sampling loss* proposed in Make-A-Shape Hui et al. (2024) and used in WaLa Sanghi et al. (2024). This loss addresses the inherent imbalance in wavelet coefficients: while coarse bands carry most of the global shape energy, the majority of detail coefficients are low in magnitude and contribute little to reconstruction when treated uniformly.

To focus the training on informative coefficients while still regularizing the rest, we first identify a set of high-magnitude entries from each detail subband $D_0$. Specifically, for each subband, we define an active set $P_0$ consisting of all spatial locations whose absolute coefficient value exceeds $\frac{1}{32}$ of the maximum in the subband $D_0$:

$$P_0 = \left\{ v \,\middle|\, |D(v)| \geq \frac{1}{32} \cdot \max_u |D(u)| \right\}. \tag{12}$$

The complement $P_0'$ contains less significant entries. To maintain efficiency, we randomly sample the same number of entries from $P_0'$ using a random sampling function $R(\cdot)$.

The total reconstruction loss is then given by:

$$\mathcal{L}_{\text{rec}} = \text{MSE}(C_0, C_0') + \frac{1}{2} \sum_{D \in \{D_0, D_1\}} \Big[ \text{MSE}(D[P_0], D'[P_0]) \\ + \text{MSE}(R(D[P_0']), R(D'[P_0'])) \Big]. \tag{13}$$

where:

- $C_0$ and $C_0'$ are the ground-truth and reconstructed coarse subbands,

- $D$ and $D'$ are the ground-truth and reconstructed detail coefficients,

- MSE denotes mean squared error over the selected spatial indices.

This loss formulation prioritizes learning from salient regions of the input while still applying light regularization to the remainder, improving training efficiency and reconstruction fidelity without introducing sampling bias.

## C  CHANNEL PERMUTATION AND NEGATION UNDER 3D ROTATIONS

In the 2D case described in the main paper, rotating an image by 90 degrees counter-clockwise results in a predictable transformation of wavelet subbands: specifically, the $HL$ and $HH$ components are negated, and the $HL$ and $LH$ subbands are swapped. This behavior arises from the inherent symmetries of biorthogonal wavelet filters, *i.e.*, `bior3.9`.

We now generalize this concept to the 3D case, where the wavelet decomposition produces eight subbands:

$$\{LLL, LLH, LHL, LHH, HLL, HLH, HHL, HHH\},$$

corresponding to the combinations of low-pass (L) and high-pass (H) filtering along the $x$, $y$, and $z$ axes. To rotate the wavelet-tree volume by 90 degrees along one of the primary axes ($x$, $y$, or $z$), we apply a combination of spatial permutation and sign flipping to the appropriate subbands.

**Negation.**  Wavelet coefficients require sign flipping when a rotation inverts the direction of a high-pass filtering axis. For example, a 90-degree clockwise rotation about the $z$-axis reverses the $y$-axis. Consequently, any subband involving high-pass filtering along the $y$-axis must be negated. In this case, we negate:

$$\{LHL, LHH, HHL, HHH\}.$$

Likewise, a 90-degree clockwise rotation about the $x$- or $y$-axis reverses the $z$-axis. Therefore, we negate all subbands containing high-pass filtering along the $z$-axis:

$$\{LLH, LHH, HLH, HHH\}.$$

Note that some subbands may be affected by multiple axes and appear in both cases, such as $LHH$ and $HHH$, which contain high-pass filtering along more than one axis.

**Permutation.**  In addition to sign flips, wavelet subbands must be permuted to reflect the new orientation of the axes after rotation. We categorize the required permutations into two types. The first type concerns reordering subbands that contain exactly one high-pass and one low-pass filter along the two non-rotated axes. For instance, a 90-degree rotation about the $x$-axis affects the $y$ and $z$ axes, requiring the following subband swaps:

$$LLH \leftrightarrow LHL, \quad HLH \leftrightarrow HHL$$

These swaps ensure that the directional semantics of the subbands remain consistent under axis permutation.

The second type of permutation arises from the reordering of the $2 \times 2 \times 2$ spatial children in the first detail level $D_1$. Because these children are stacked channel-wise in our implementation, a 90-degree rotation induces a spatial permutation of their voxel indices. For example:

- Rotation around the $x$-axis: $(x, y, z) \rightarrow (x, -z, y)$
- Rotation around the $y$-axis: $(x, y, z) \rightarrow (-z, y, x)$
- Rotation around the $z$-axis: $(x, y, z) \rightarrow (-y, x, z)$

Applying these coordinate transformations to the children of a given subband yields the required channel reordering. Both types of permutations, subband swaps and intra-subband voxel reordering, can be implemented efficiently using precomputed lookup tables, one for each rotation in the octahedral group $O$.

## D  LIMITATION AND FUTURE WORK

While our framework demonstrates strong performance, several limitations and extensions remain. First, although we adopt wavelet-tree volumes for their regularity and rotational structure, applying our equivariant approach to more flexible representations like vector sets Zhang et al. (2023) may offer improved performance. Second, while we currently focus on equivariance to discrete 90-degree rotations, extending the framework to continuous rotational equivariance could allow finer-grained modeling of orientation variability, which may further improve generative performance in practice. Finally, scaling beyond the current $256^3$ resolution to $512^3$ or $1024^3$ using hierarchical strategies could further enhance shape fidelity.

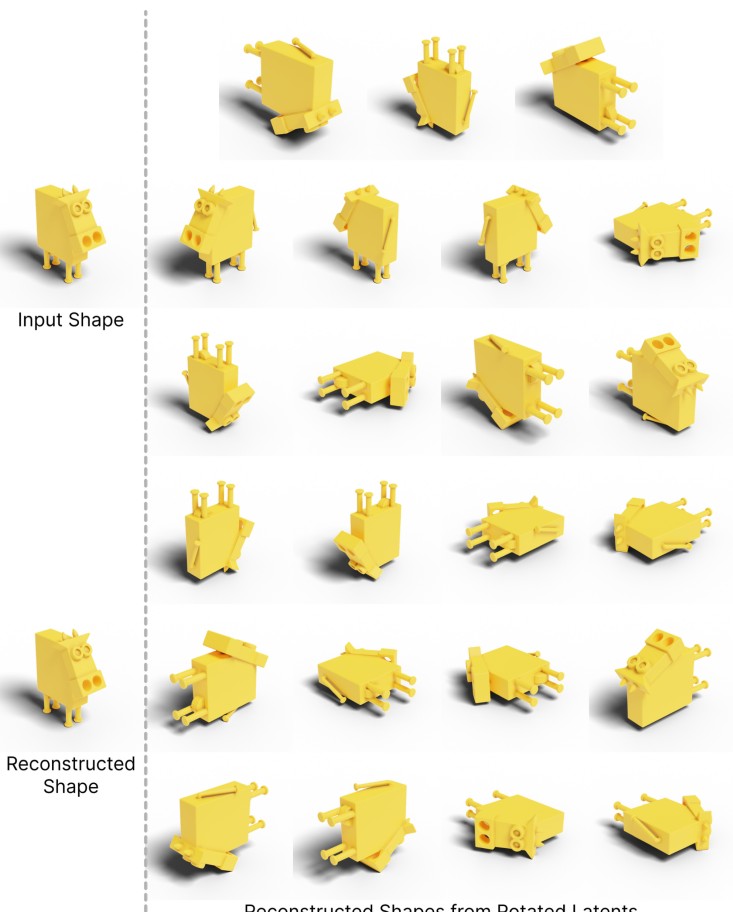

Input Shape

Reconstructed
Shape

Reconstructed Shapes from Rotated Latents

Figure 7: Latent space equivariance results. Starting from the input shape (top left), our autoencoder accurately reconstructs it (bottom left). Applying discrete 3D rotations directly to the latent code and decoding yields shapes in the matching rotated orientations (right), demonstrating consistent latent–space equivariance.

# E    EQUIVARIANT REGULARIZED LATENT SPACE VISUALIZATION

To verify the effectiveness of our equivariant regularization, we visualize reconstructions from rotated latent codes in Figure 7. The input shape is first encoded into its latent representation, which is then rotated by discrete elements of the octahedral group before decoding. The resulting reconstructions exhibit consistent geometry and correct relative orientations across all rotations, closely matching the expected transformations of the original shape. This behavior empirically confirms that the latent space of our model is approximately equivariant, ensuring that rotation in the latent domain corresponds to rotation in the reconstructed geometry.

# F    OBJAVERSE UNCONDITIONAL GENERATION RESULTS

We further evaluate our framework on the Objaverse dataset to assess its scalability and generalization in large-scale, diverse 3D domains. The dataset encompasses a broad spectrum of object types with high geometric and semantic complexity, making unconditional generation particularly challenging. As shown in Figure 8, our model successfully synthesizes coherent and unoriented 3D shapes across varied categories, including organic, mechanical, and artistic forms. The generated samples exhibit high visual fidelity and structural diversity, demonstrating that our framework maintains robustness and quality even when trained on large datasets.

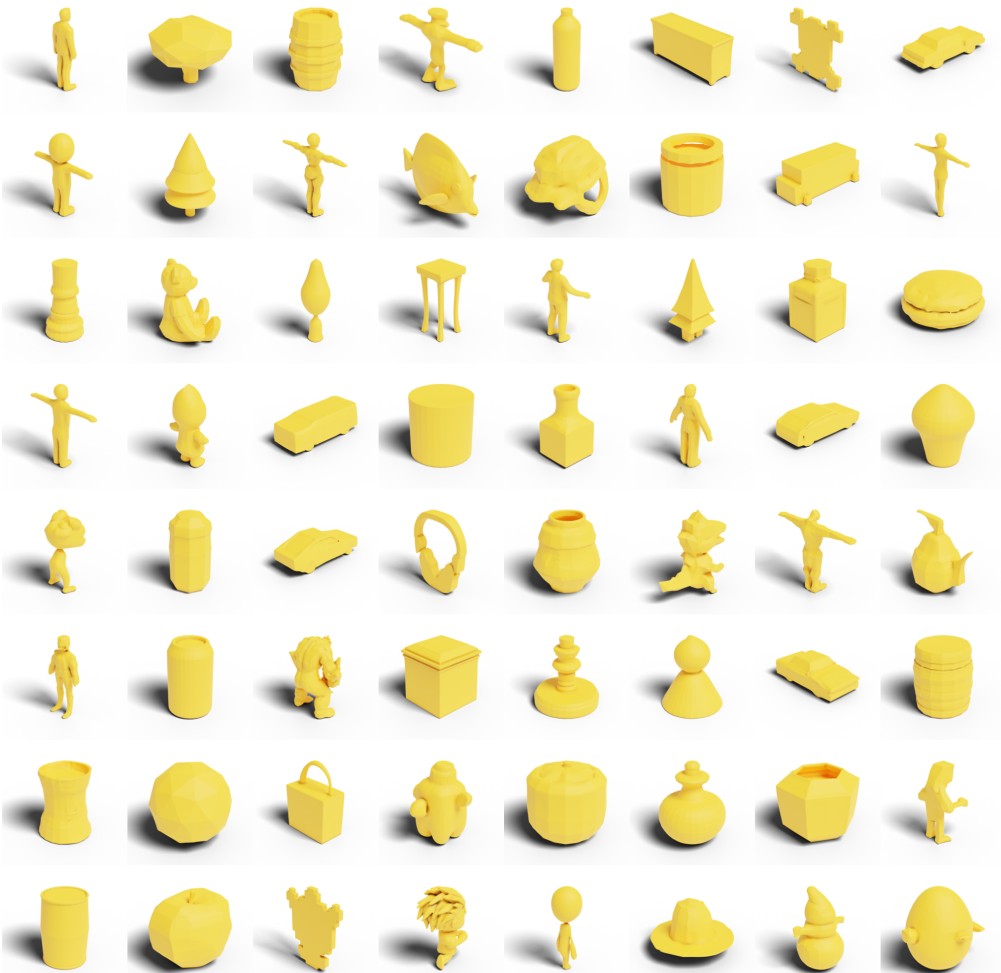

Figure 8: Our framework generalizes well to large-scale datasets such as Objaverse Deitke et al. (2023; 2022), generating 3D objects with diverse categories and varying orientations.

## G   TEXT-CONDITIONED GENERATION RESULTS

**Setting.** Text-conditioned 3D generation evaluates a model's ability to produce shapes that are not only diverse and high-quality but also semantically aligned with language prompts. To support this, we modify our DiT-based generator by replacing self-attention layers with cross-attention, allowing it to condition on text inputs. We encode text using a pretrained CLIP text encoder Radford et al. (2021), and inject the resulting embeddings into the generator to guide shape synthesis based on caption semantics. We train our models using the filtered Objaverse subset provided by TRELLIS Xiang et al. (2024), which curates high-quality, captioned 3D shapes for language-guided generation.

**Visual Gallery.** We present a curated visual gallery of our text-conditioned generation results in Figure 9, showcasing the effectiveness of our method across a wide range of language prompts. The generated shapes demonstrate strong alignment with their corresponding captions, successfully capturing key semantic attributes such as object category, structure, and geometric features. Our model consistently produces shapes with fine-grained details, sharp boundaries, and plausible proportions, reflecting both the diversity and realism present in the training dataset. These qualitative examples affirm the capability of our approach to perform high-quality, language-guided 3D synthesis, supporting both visual plausibility and caption alignment.

**Evaluation Metrics.** We evaluate text-conditioned 3D generation using both visual quality and language alignment metrics. Following TRELLIS Xiang et al. (2024), we report Fréchet Inception Distance (FID) and Kernel Inception Distance (KID) computed on rendered images of generated

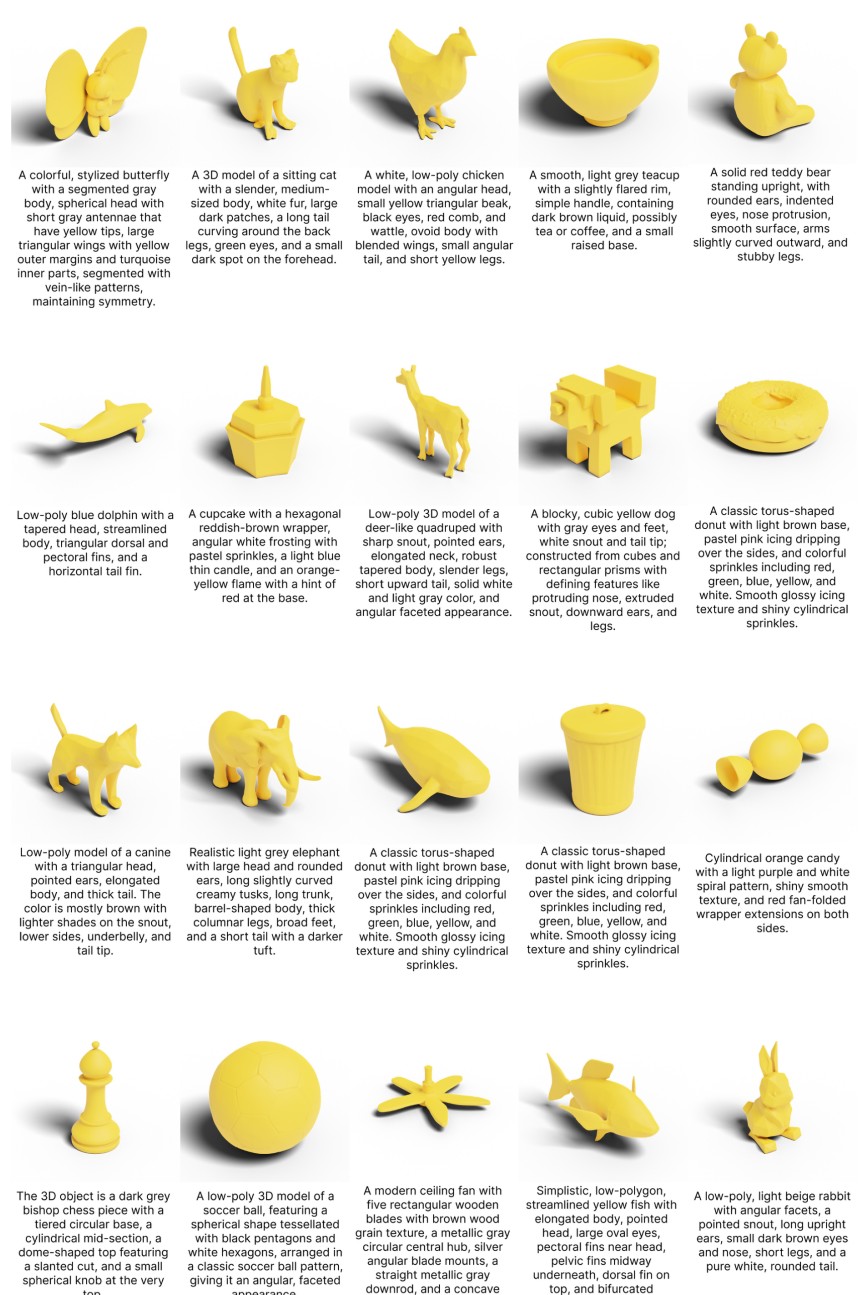

Figure 9: Example text-to-shape generations from Objaverse Deitke et al. (2022; 2023). Our model produces diverse and semantically aligned 3D objects, including animals (e.g., cat, chicken, dolphin), food items (e.g., donuts, cupcake), toys, and household objects (e.g., teacup, fan, chess piece), faithfully reflecting the input prompts.

shapes to assess overall visual fidelity and realism. Lower FID and KID values indicate better perceptual quality and distributional similarity to real shapes. To measure text-shape alignment, we use CLIP-S, which computes the cosine similarity between the CLIP image embeddings of rendered shapes and the CLIP text embeddings of the corresponding input prompts. Higher CLIP-S scores indicate stronger semantic consistency between the generated shape and the guiding caption.

**Quantitative Comparisons.** We compare our method against recent state-of-the-art approaches for text-to-3D generation, including XCube Zheng et al. (2023) and TRELLIS Xiang et al. (2024). Results are summarized in Table 3. Our model achieves the comparable Fréchet Inception Distance (FID) and Kernel Inception Distance (KID) with TRELLIS and significantly outperforms Xcube,

Table 3: Comparison of text-to-3D generation performance. Our method achieves competitive visual quality, reflected in comparable or lower FID and KID, while significantly improving text-shape alignment, as indicated by a higher CLIP-S score.

| Method | FID ↓ | KID ↓ | CLIP-S ↑ |
|---|---|---|---|
| XCube Zheng et al. (2023) | 57.55 | 2.923 | 18.33 |
| TRELLIS Xiang et al. (2024) | **16.23** | **0.354** | 21.05 |
| Ours | 21.67 | 0.724 | **21.62** |

Low-poly deer with cylindrical body, angular head, pointed antlers, cylindrical neck, segmented legs (with hooves), and short conical tail.

A butterfly with a small, rounded head, segmented thorax, cylindrical abdomen, large forewings, smaller hindwings, eye spots near the hindwing edges.

A stylized bunny rabbit with a rounded head and body, elongated oval-shaped ears, cylindrical arms and legs, arms angled downward and forward, and legs bent suggesting a seated posture

A children's tricycle with handlebars and grips, a frame, a polygonal seat, a large front wheel with pedals, smaller rear wheels, and a footrest.

A single-engine propeller airplane with mid-body rectangular wings, a vertical tail fin, two horizontal stabilizers, a bubble canopy, and tricycle landing gear.

Figure 10: Qualitative text-conditioned generation comparisons. For each prompt, our method produces clean, structurally coherent shapes that match the intended semantics, achieving quality comparable to TRELLIS Xiang et al. (2024) while significantly outperforming XCube Ren et al. (2024), which often yields incomplete or distorted geometry. Examples include a deer with well-defined antlers and limbs, a butterfly with distinct wing anatomy, a rounded bunny figure, a tricycle with correct topology, and an airplane with clearly formed wings.

indicating that the generated shapes are visually realistic and well-aligned with the target distribution of real 3D shapes. More notably, our method attains the highest CLIP-S score, demonstrating better semantic consistency between the generated 3D shapes and their associated text prompts. Overall, these results validate the effectiveness of our approach for high-fidelity, semantically grounded 3D generation guided by natural language.

**Qualitative Comparisons.** Figure 10 presents qualitative comparisons with TRELLIS Xiang et al. (2024) and XCube Zheng et al. (2023) across several representative text prompts. Our method produces clean, coherent, and semantically faithful shapes, often matching the visual quality of TRELLIS while exhibiting substantially better geometric stability and structural completeness than XCube. In contrast, XCube frequently generates distorted or incomplete geometry, missing limbs, collapsed surfaces, or implausible proportions, while TRELLIS and our approach both maintain consistent object structure. Overall, these comparisons show that our model achieves competitive text-conditioned quality with TRELLIS and significantly surpasses XCube in geometric fidelity and prompt alignment.

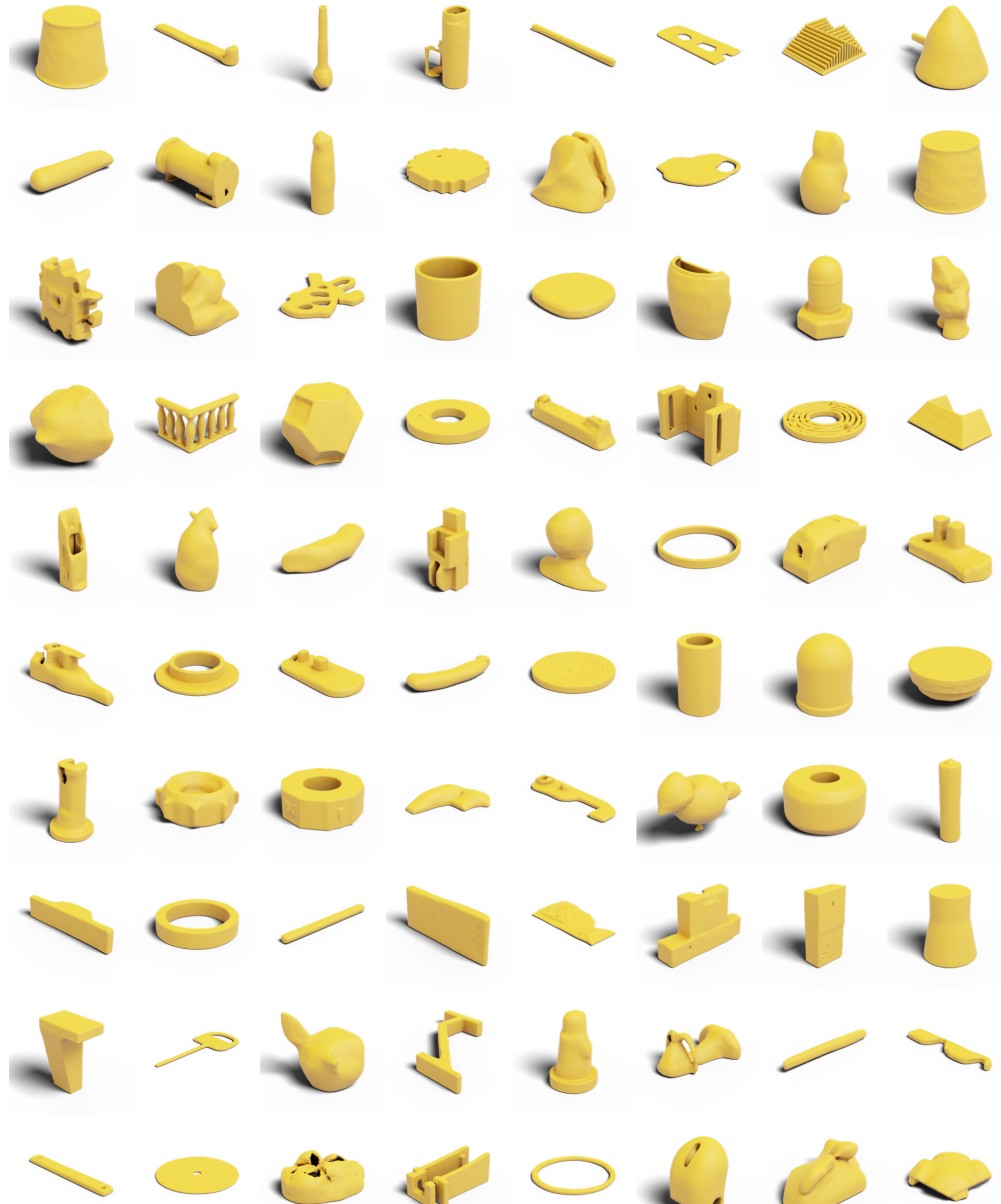

Figure 11: Random Generation results produced by LAS-Diffusion Zheng et al. (2023).

## H UNCONDITIONAL QUALITATIVE COMPARISONS

We provide the quantitative comparisons of unconditional generation in Figures 11- 19. Our results demonstrate that the proposed framework consistently produces more coherent global structures, sharper surface details, and higher geometric diversity compared to the baselines.

## I ADDITIONAL AUTOENCODING RESULTS

Additional autoencoding comparisons are provided in Figure 20. Our model faithfully recovers fine details and complex geometry, closely matching the ground truth. In contrast, baseline methods often exhibit artifacts such as structural collapse, excessive smoothing, or topological errors. These results demonstrate the superior reconstruction fidelity and structural consistency achieved by our framework.

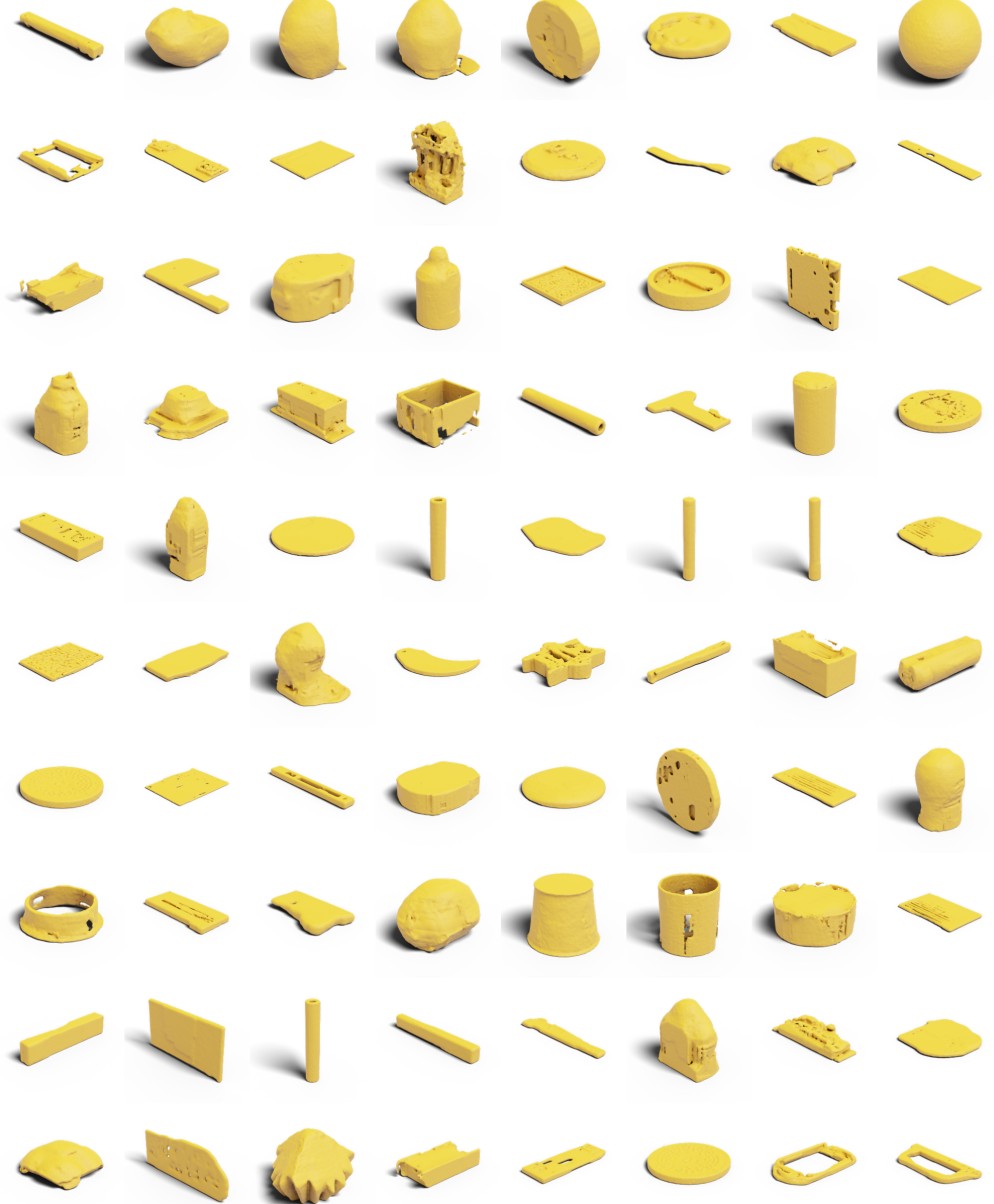

Figure 12: Random Generation results produced by OctFusion Xiong et al. (2024).

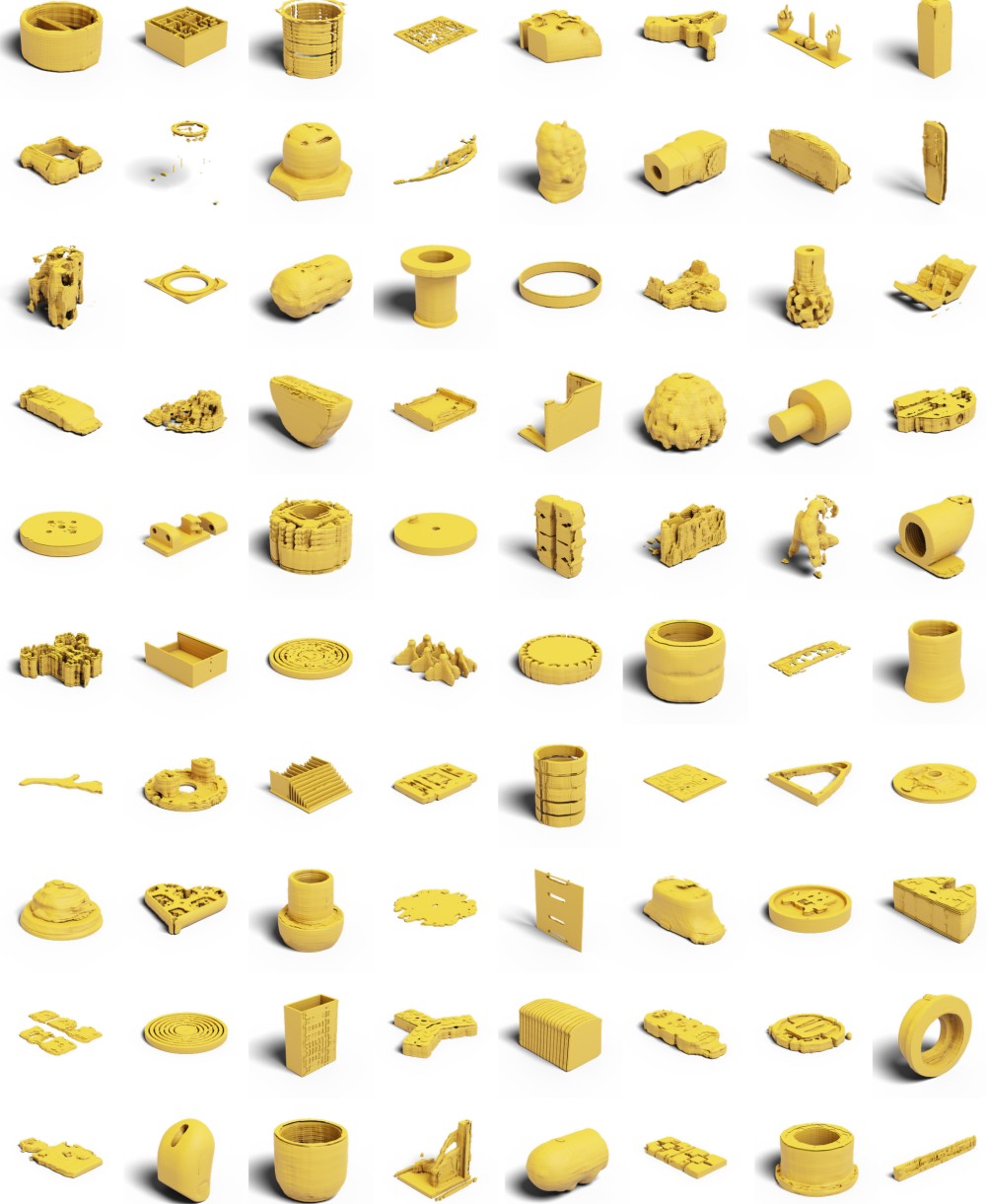

Figure 13: Random Generation results produced by Shape2Vecset Zhang et al. (2023).

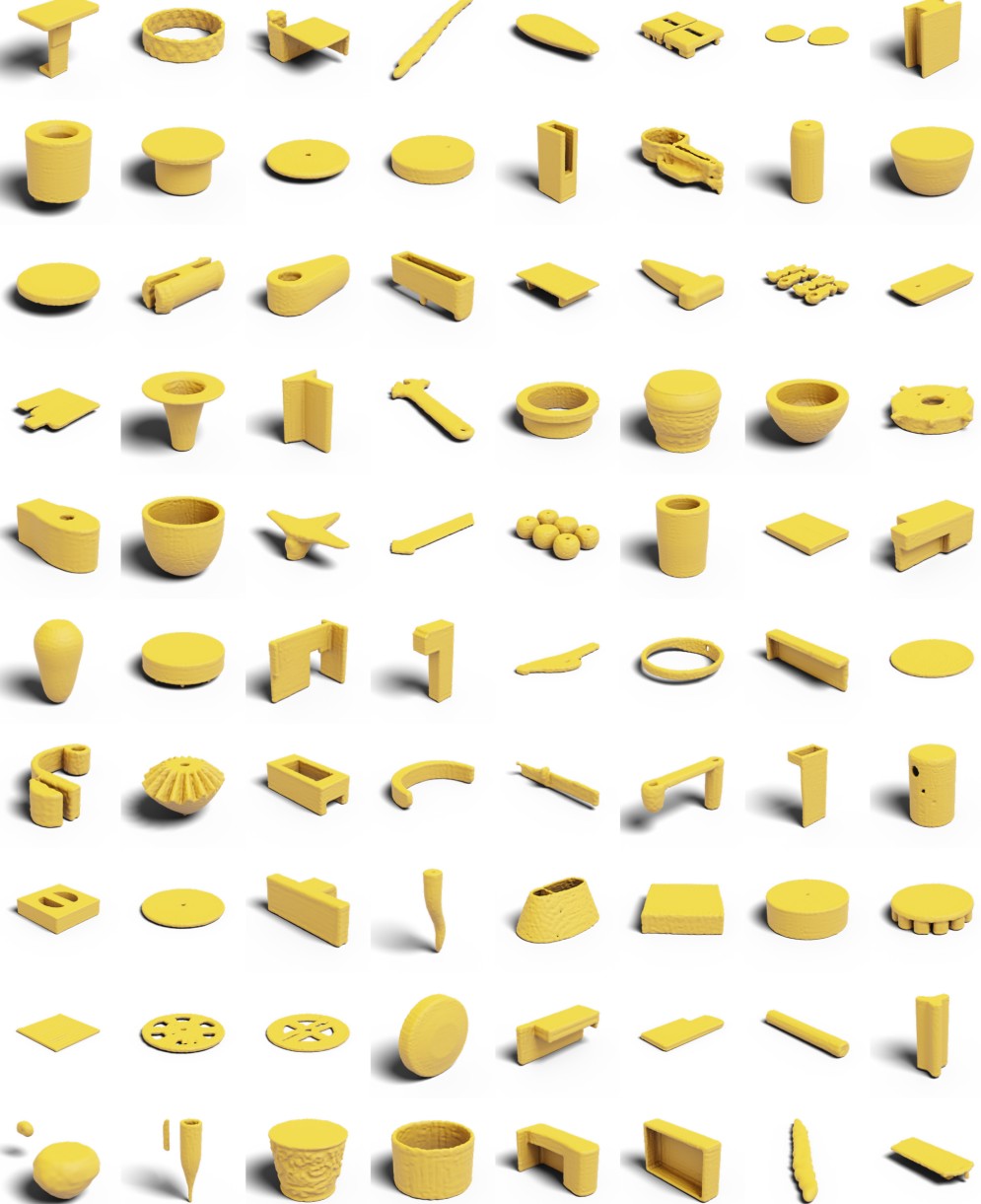

Figure 14: Random Generation results produced by WaveGen Hui et al. (2022).

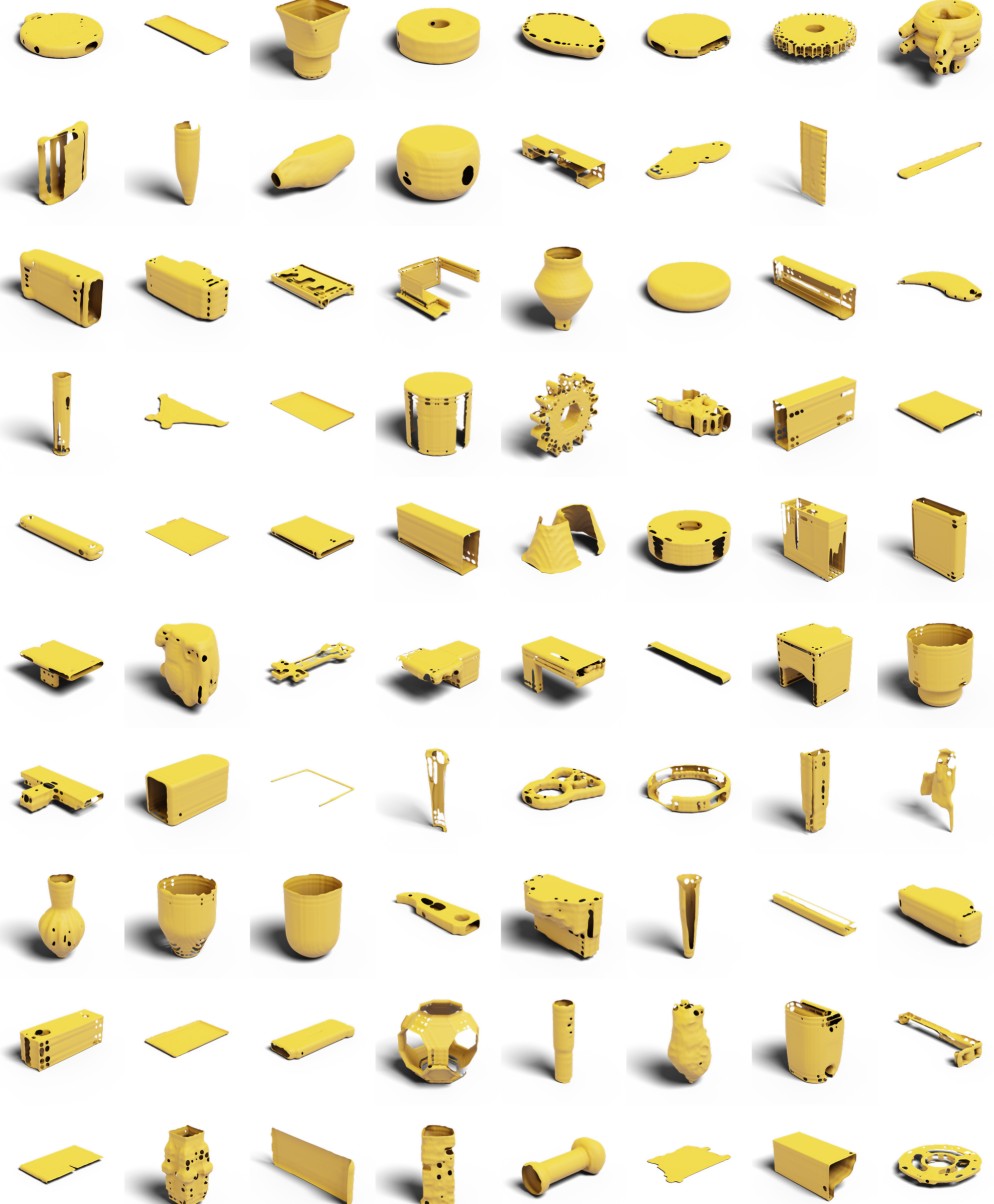

Figure 15: Random Generation results produced by UDiff Zhou et al. (2024).

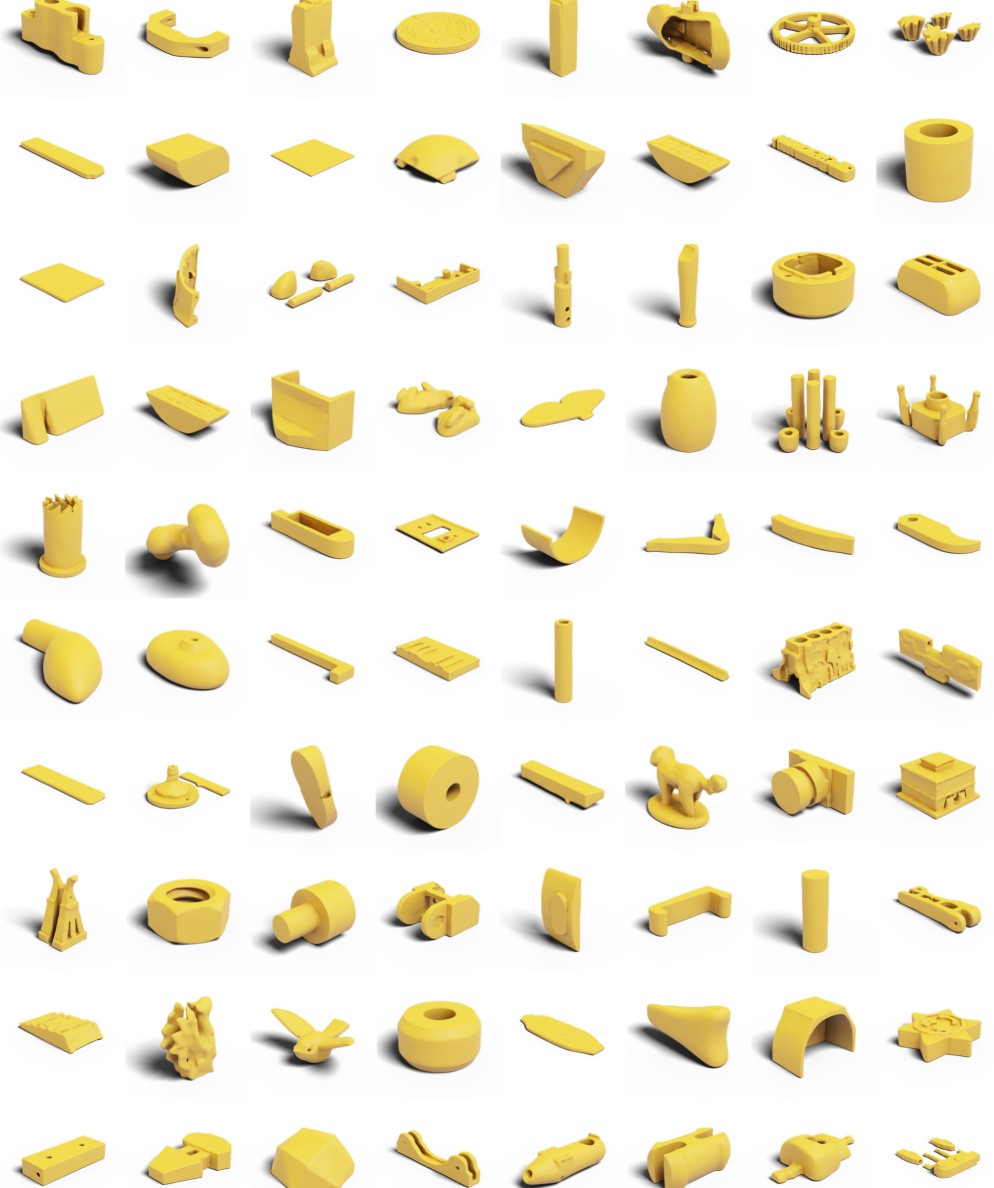

Figure 16: Random Generation results produced by WaLa Sanghi et al. (2024).

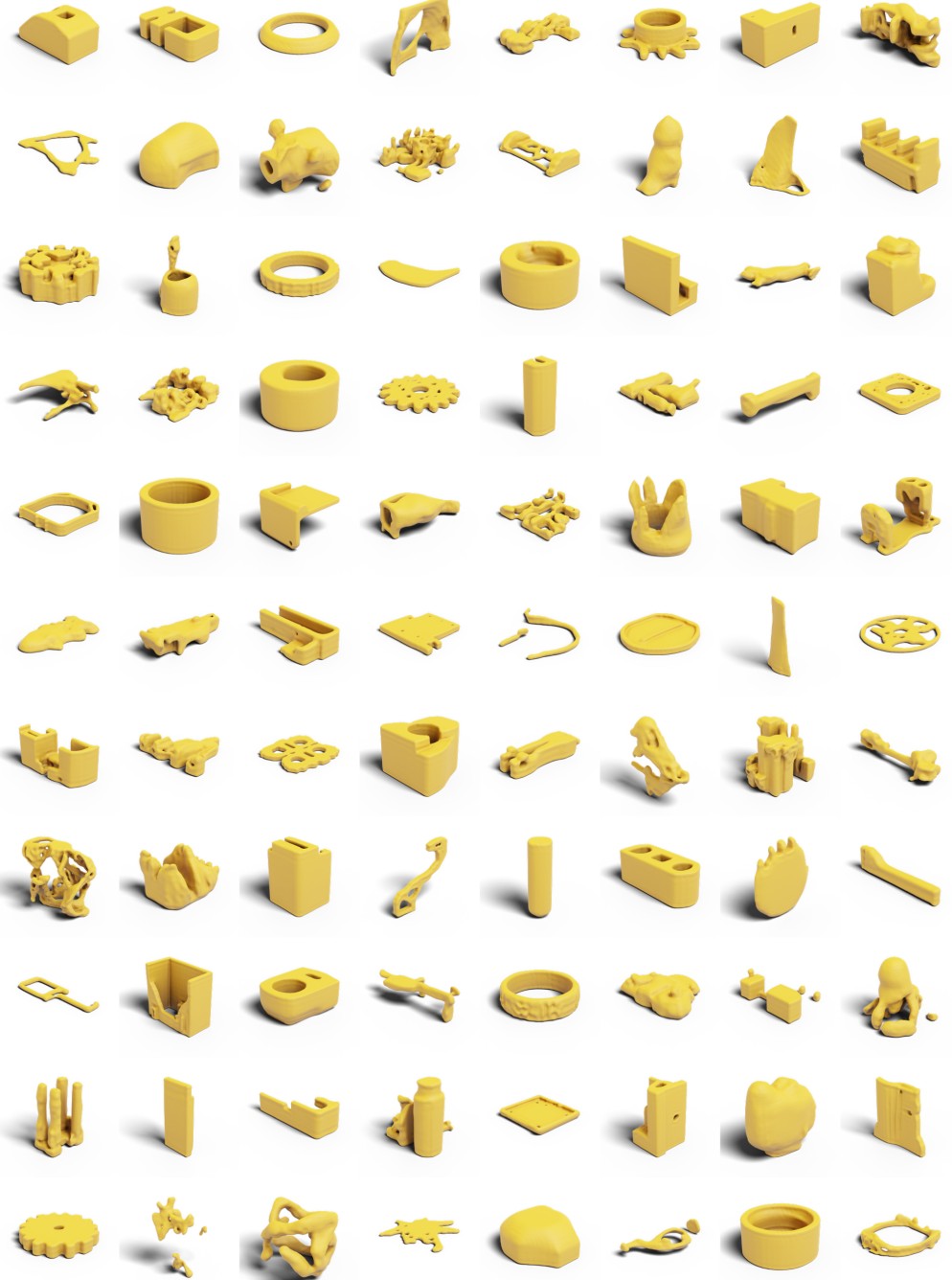

Figure 17: Random Generation results produced by Make-A-Shape Hui et al. (2024).

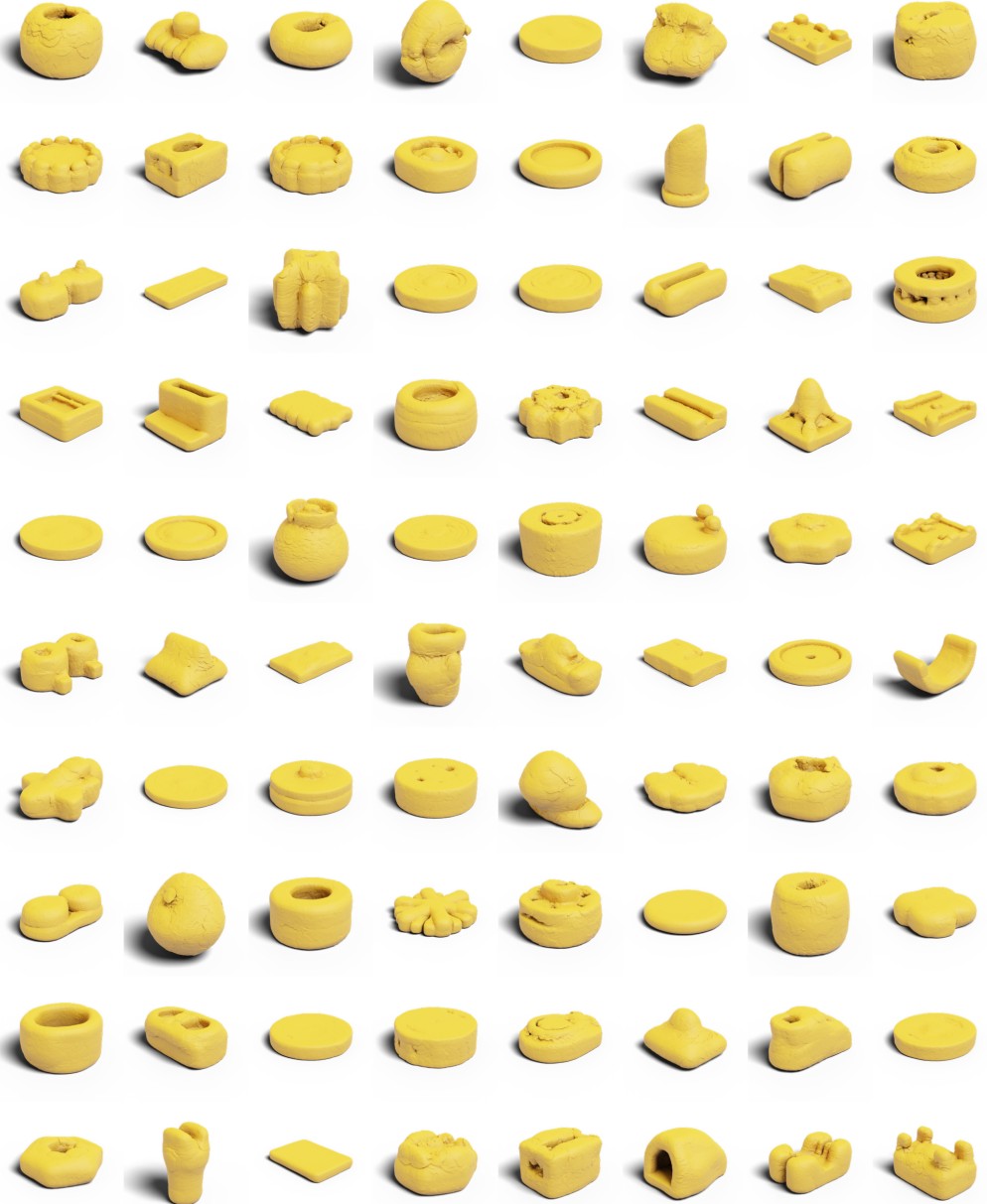

Figure 18: Random Generation results produced by GEM3D Petrov et al. (2024).

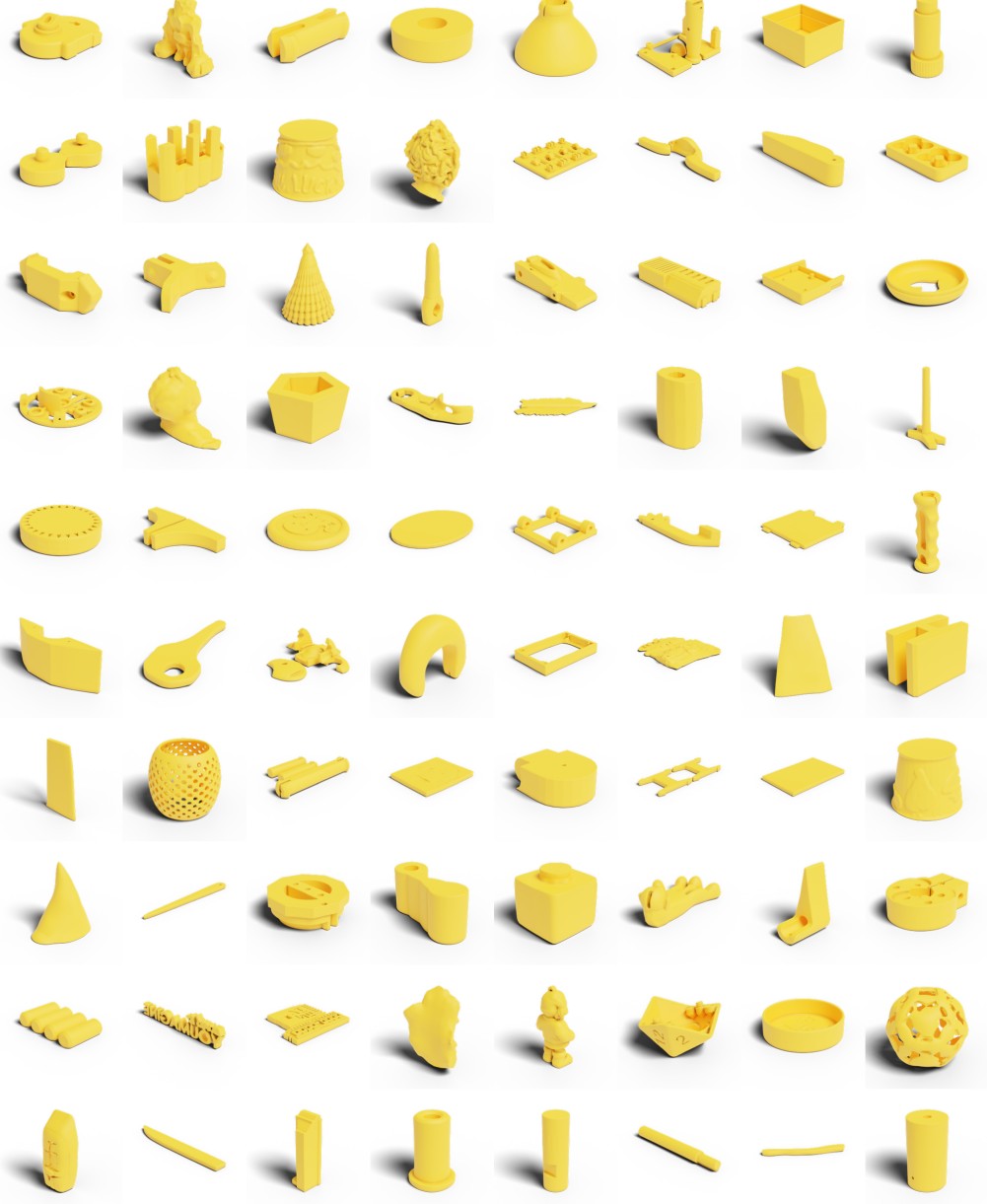

Figure 19: Random Generation results produced by our framework.

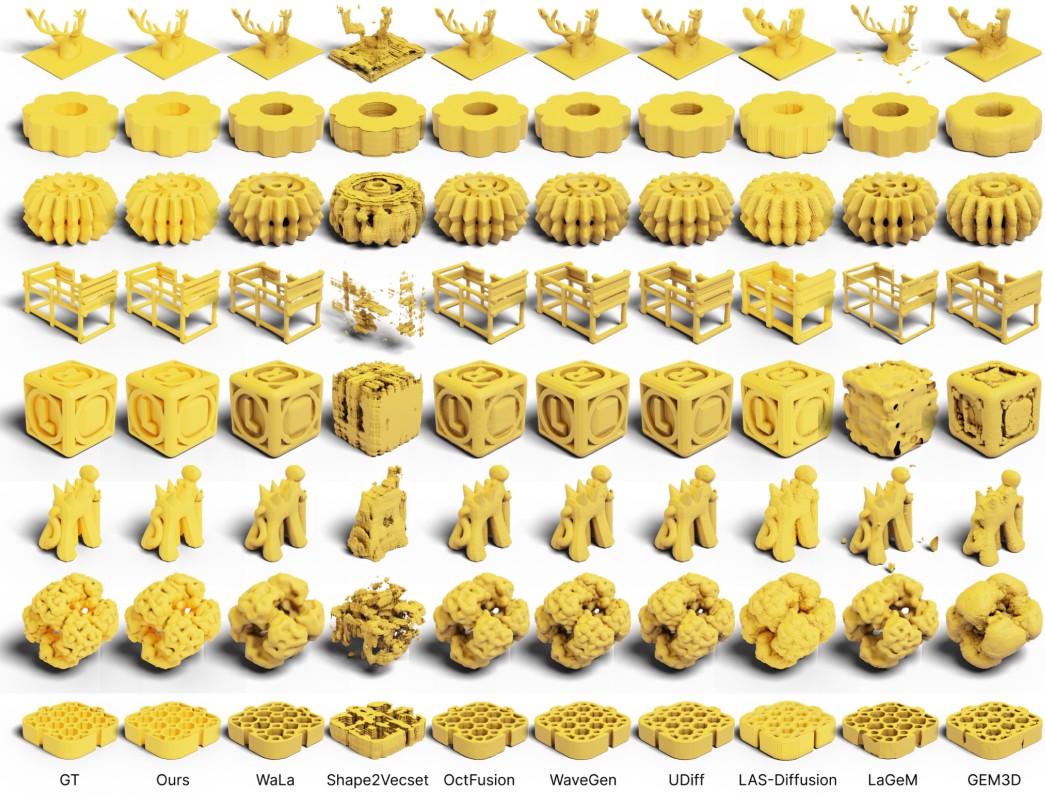

Figure 20: Additional qualitative comparison of autoencoded 3D shapes. Our method accurately preserves structure and fine details, while baselines often suffer from smoothing, collapse, or topological errors.

