# OpenReview forum: "Eq-WaLa: Equivariant Augmentation and Regularization for Wavelet Latent Flow Matching"
_ICLR.cc/2026/Workshop/GRaM — ICLR 2026 Workshop GRaM Poster_

### Official Review · Reviewer_MNNP · 2026-02-19
**EQ-WALA**

**Rating:** 5
**Confidence:** 3

**Review:**

## Summary
The paper proposes EQ-WALA, an equivariant generative framework designed to address the sensitivity of latent implicit 3D models to arbitrary coordinate frames, particularly in unaligned datasets. The method builds upon a compact wavelet-tree representation of Signed Distance Fields (SDFs). To explicitly account for rotational variability, the authors introduce two key technical contributions:
- Wavelet-Domain Rotation Augmentation: A novel scheme that constructs rotated inputs on-the-fly by permuting and negating wavelet coefficients, leveraging the inherent symmetries of the wavelet transform.
- Equivariant Latent Regularization: A training strategy that enforces consistency between the encodings of rotated shapes and the rotated latents of original shapes.

These components are integrated into a Flow Matching generative model trained on the regularized latent space. The framework is evaluated on the unaligned Thingi10K dataset and the large-scale Objaverse dataset, demonstrating improved reconstruction accuracy, faster convergence, and scalable text-conditioned generation

## Strengths
- **Efficient On-the-Fly Augmentation:** The proposed wavelet-domain augmentation is mathematically exact and computationally efficient. By performing lightweight tensor operations (permutation and sign flipping) directly on coefficients, it avoids the overhead of preprocessing or storing rotated volumes, effectively multiplying dataset size with negligible cost.
- **State-of-the-Art Performance on Unaligned Data:** The method demonstrates significant improvements on the Thingi10K dataset, which lacks canonical orientation. It achieves higher Intersection over Union (IoU) and lower Light Field Distance (LFD) compared to baselines like WaLa and OctFusion.
- **Accelerated Generative Convergence:** The inclusion of equivariant regularization not only improves final quality but also speeds up training convergence by approximately 1.6x (reaching comparable FID at 240k iterations vs. 400k for the baseline).
- **Scalability and Multimodality:** The authors successfully scale the framework to the massive Objaverse dataset and demonstrate text-conditioned generation. This proves the method's robustness for complex, real-world generative tasks beyond simple unconditional synthesis

## **Weaknesses**

- **Lack of Detail on the `bior3.9` Wavelet Basis:** The authors identify the `bior3.9` wavelet filter as a critical component of their "Wavelet Domain Rotation Augmentation," stating that "certain symmetry properties emerge" when this basis is used. However, the paper never actually defines what the `bior3.9` filters are. There is no presentation of the filter coefficients (taps), no specific mathematical definition of their symmetries, and no formal explanation of *why* this specific biorthogonal basis guarantees exact 90-degree rotational equivariance through simple channel permutation and negation.
- **Relation to Group Equivariant CNNs (G-CNNs):** The paper claims it does "not enforce strict architectural equivariance", yet the method of augmenting inputs and enforcing feature map consistency is conceptually similar to the "lifting" and "projection" operations defined in foundational G-CNN literature. The authors fail to discuss why a soft regularization loss was chosen over the hard architectural constraints found in **[1]**, or more modern 3D equivariant architectures like SE(3)-Transformers **[2]** or Vector Neurons **[3]**, which guarantee equivariance without tuning loss weights.
- **Rediscovery of Wavelet Steerability:** The "key observation" that 90-degree rotations correspond to subband permutation and negation is a known property in signal processing, often referred to as "steerability". The derivation is presented as a novel finding rather than an application of standard separable wavelet theory, missing connections to classic works like Steerable Pyramids **[4]**, which explicitly established the conditions for rotation-invariant wavelet design.
- **Parallels with Contrastive Learning:** The equivariant regularization loss, which minimizes the distance between rotated latents and encodings of rotated inputs, mirrors objectives in self-supervised representation learning. Specifically, it resembles SimCLR **[5]** and its rotation-aware variants like Equivariant Contrastive Learning **[6]**. Framing this purely as a "generative regularization" strategy ignores existing literature.

## **Questions**

1.  **Architectural Choice:** Why did you opt for a "soft" regularization loss rather than utilizing an architecture that guarantees equivariance by design (e.g., G-CNNs **[1]**, SE(3)-Transformers **[2]**, or Vector Neurons **[3]**), given that the latter would enforce the property exactly without a loss weight hyperparameter?
2.  **Canonicalization:** Did you consider or compare against a baseline that uses a canonicalization module to learn to rotate inputs to a standard frame (such as Spatial Transformer Networks **[7]** or T-Net **[8]**) instead of enforcing equivariance? If not, why is equivariance preferred for this specific generative task?
3.  **Loss Sensitivity:** The training objective includes multiple weighted terms. How sensitive is the model's performance to the relative weighting of the equivariant regularization term versus the reconstruction terms, especially in light of the contrastive learning dynamics discussed in **[5]** and **[6]**?
4.  **Sampling Details:** Could you clarify the distribution used for the "random sampling" of the group elements and the wavelet coefficients in the adaptive loss? Is it uniform, or is there an importance sampling scheme?

---

### **References**

* **[1]** Cohen, T., & Welling, M. (2016). Group Equivariant Convolutional Networks. *International Conference on Machine Learning (ICML)*.
* **[2]** Fuchs, F. B., Worrall, D. E., Fischer, V., & Welling, M. (2020). SE(3)-Transformers: 3D Roto-Translation Equivariant Attention Networks. *Advances in Neural Information Processing Systems (NeurIPS)*.
* **[3]** Deng, C., Litany, O., Duan, Y., Poulenard, A., Tagliasacchi, A., & Guibas, L. J. (2021). Vector Neurons: A General Framework for SO(3)-Equivariant Networks. *IEEE International Conference on Computer Vision (ICCV)*.
* **[4]** Simoncelli, E. P., & Freeman, W. T. (1995). The Steerable Pyramid: A flexible architecture for multi-scale derivative computation. *International Conference on Image Processing (ICIP)*.
* **[5]** Chen, T., Kornblith, S., Norouzi, M., & Hinton, G. (2020). A simple framework for contrastive learning of visual representations. *International Conference on Machine Learning (ICML)*.
* **[6]** Dangovski, R., Jing, L., Loh, C., Han, S., Srivastava, A., Cheung, B., ... & Soljačić, M. (2022). Equivariant Contrastive Learning. *International Conference on Learning Representations (ICLR)*.
* **[7]** Jaderberg, M., Simonyan, K., Zisserman, A., & Kavukcuoglu, K. (2015). Spatial Transformer Networks. *Advances in Neural Information Processing Systems (NeurIPS)*.
* **[8]** Qi, C. R., Su, H., Mo, K., & Guibas, L. J. (2017). PointNet: Deep Learning on Point Sets for 3D Classification and Segmentation. *IEEE Conference on Computer Vision and Pattern Recognition (CVPR)*.

**Pmlr Suitability:**

Yes

---

### Official Review · Reviewer_5WVr · 2026-02-24
**Review Eq-WaLa**

**Rating:** 6
**Confidence:** 3

**Review:**

## Summary.

The paper tackles unaligned 3D shape generation using a latent pipeline built on wavelet-tree TSDF volumes. The core idea is to exploit discrete rotation structure in the wavelet representation so the model can apply all 90-degree rotations on-the-fly via cheap channel permutations and sign flips, rather than storing many rotated copies of the data. On top of this, it adds a latent-space equivariance regularizer: the decoder is trained to produce consistent outputs when latents are rotated, encouraging rotation-consistent reconstructions and improving downstream generative training. A DiT-style latent flow-matching generator is then trained on these latents, with reported gains on unaligned benchmarks and demonstrations at larger scale.

## Strengths
- Efficient augmentation: The rotation augmentation in wavelet space is simple and fast (permutation/sign operations), enabling full discrete rotation coverage without dataset bloat.
- Practical equivariance without specialized networks: Encourages rotation-consistent behavior through a regularizer rather than requiring fully equivariant architectures, which can be harder to scale.
- Strong empirical improvements: Reports clear gains over wavelet-latent baselines on reconstruction and unconditional generation on unaligned data.
- Clean pipeline: The method is conceptually straightforward and easy to integrate into existing latent generative setups.

## Weaknesses
- Narrow symmetry scope: Only handles 90-degree rotations; it’s unclear how well the approach extends to arbitrary rotations, which is the more common real-world issue.
- Incremental novelty: Much of the contribution is careful engineering around known wavelet rotation behavior plus a fairly standard consistency loss.
- Heavy compute footprint: The reported training budgets are very large, raising reproducibility concerns and making it hard to judge cost–benefit under typical resources.
- Equivariance not fully stress-tested: Evaluation focuses on reconstruction/generation metrics; there’s limited direct measurement of rotation-consistency errors or robustness under systematic rotation perturbations.
- Comparisons not fully comprehensive: Some relevant baselines are missing or only partially evaluated (e.g., due to code availability), weakening the strength of the empirical claim.

**Pmlr Suitability:**

Yes

---

### Meta-Review · Area_Chair_PkMi · 2026-02-28

**Decision:**

Accept

**Metareview:**

I recommend an accept and suggest the authors to answer the concerns raised by the reviewers for the camera ready version.

**Relevance To Proceedings:**

Yes — suitable for PMLR (long paper)

**Relevance To Workshop:**

Yes — suitable for GRaM

---

### Decision · Program_Chairs · 2026-03-02

Accept (Poster)